# Personalized Management of Malignant and Non-Malignant Ectopic Mediastinal Thyroid: A Proposed 10-Item Algorithm Approach

**DOI:** 10.3390/cancers16101868

**Published:** 2024-05-14

**Authors:** Mara Carsote, Mihai-Lucian Ciobica, Oana-Claudia Sima, Adrian Ciuche, Ovidiu Popa-Velea, Mihaela Stanciu, Florina Ligia Popa, Claudiu Nistor

**Affiliations:** 1Department of Endocrinology, “Carol Davila” University of Medicine and Pharmacy, 020021 Bucharest, Romania; carsote_m@hotmail.com; 2Department of Clinical Endocrinology V, “C.I. Parhon” National Institute of Endocrinology, 011863 Bucharest, Romania; 3Department of Internal Medicine and Gastroenterology, “Carol Davila” University of Medicine and Pharmacy, 020021 Bucharest, Romania; 4Department of Internal Medicine I and Rheumatology, “Dr. Carol Davila” Central Military University Emergency Hospital, 010825 Bucharest, Romania; 5PhD Doctoral School, “Carol Davila” University of Medicine and Pharmacy, 020021 Bucharest, Romania; oana-claudia.sima@drd.umfcd.ro; 6Department 4-Cardio-Thoracic Pathology, Thoracic Surgery II Discipline, “Carol Davila” University of Medicine and Pharmacy, 050474 Bucharest, Romania; claudiu.nistor@umfcd.ro; 7Thoracic Surgery Department, “Dr. Carol Davila” Central Military University Emergency Hospital, 010242 Bucharest, Romania; 8Department of Medical Psychology, Faculty of Medicine, Carol Davila University of Medicine and Pharmacy, 050474 Bucharest, Romania; ovidiu.popa-velea@umfcd.ro; 9Department of Endocrinology, Faculty of Medicine, Lucian Blaga University of Sibiu, 550024 Sibiu, Romania; mihaela.stanciu@ulbsibiu.ro; 10Department of Physical Medicine and Rehabilitation, Faculty of Medicine, “Lucian Blaga” University of Sibiu, 550024 Sibiu, Romania; florina-ligia.popa@ulbsibiu.ro

**Keywords:** thyroid cancer, dyspnea, thyroidectomy, mediastinum, malignancy, ectopic, VATS, thoracic surgery, fine needle aspiration, biopsy

## Abstract

**Simple Summary:**

A large body of multidisciplinary evidence involves the topic of thyroid cancer (the most common endocrine malignancy). Nevertheless, exceptional findings such as thyroid cancer in ectopic thyroid tissue, representing 0.3–0.5% of the malignant neoplasia with any location, suggest even greater challenges. Awareness remains the key operative element since the index of suspicion is low, especially in non-cervical areas. Hence, currently, the ectopic thyroid remains a matter of individualized management. The ectopic mediastinal thyroid (EMT) is part of the less frequent sublingual ectopic sites. Here, we introduce the most complex analysis in published EMT data (N = 117 patients) that identified an unexpectedly high rate of malignancy (18.8%), papillary cancer being the most frequent histological type. A rate of 5.98% amid all EMTs represented individuals confirmed with unrelated (non-thyroid) malignancies. Thyroid anomalies (other than EMT presence) were reported in 38.33% of the benign EMT, while the overall malignancy rate in EMTs was higher than expected according to prior data when compared to other ectopic sites.

**Abstract:**

We aimed to analyze the management of the ectopic mediastinal thyroid (EMT) with respect to EMT-related cancer and non-malignant findings related to the pathological report, clinical presentation, imaging traits, endocrine profile, connective tissue to the cervical (eutopic) thyroid gland, biopsy or fine needle aspiration (FNA) results, surgical techniques and post-operatory outcome. This was a comprehensive review based on revising any type of freely PubMed-accessible English, full-length original papers including the keywords “ectopic thyroid” and “mediastinum” from inception until March 2024. We included 89 original articles that specified EMTs data. We classified them into four main groups: (I) studies/case series (n = 10; N = 36 EMT patients); (II) malignant EMTs (N = 22 subjects; except for one newborn with immature teratoma in the EMT, only adults were reported; mean age of 62.94 years; ranges: 34 to 90 years; female to male ratio of 0.9). Histological analysis in adults showed the following: papillary (N = 11/21); follicular variant of the papillary type (N = 2/21); Hürthle cell thyroid follicular malignancy (N = 1/21); poorly differentiated (N = 1/21); anaplastic (N = 2/21); medullary (N = 1/21); lymphoma (N = 2/21); and MALT (mucosa-associated lymphoid tissue) (N = 1/21); (III) benign EMTs with no thyroid anomalies (N = 37 subjects; mean age of 56.32 years; ranges: 30 to 80 years; female to male ratio of 1.8); (IV) benign EMTs with thyroid anomalies (N = 23; female to male ratio of 5.6; average age of 52.1 years). This panel involved clinical/subclinical hypothyroidism (iatrogenic, congenital, thyroiditis-induced, and transitory type upon EMT removal); thyrotoxicosis (including autonomous activity in EMTs that suppressed eutopic gland); autoimmune thyroiditis/Graves’s disease; nodules/multinodular goiter and cancer in eutopic thyroid or prior thyroidectomy (before EMT detection). We propose a 10-item algorithm that might help navigate through the EMT domain. To conclude, across this focused-sample analysis (to our knowledge, the largest of its kind) of EMTs, the EMT clinical index of suspicion remains low; a higher rate of cancer is reported than prior data (18.8%), incident imagery-based detection was found in 10–14% of the EMTs; surgery offered an overall good outcome. A wide range of imagery, biopsy/FNA and surgical procedures is part of an otherwise complex personalized management.

## 1. Introduction

Thyroid cancer represents the most common endocrine malignancy. For the majority, there are papillary and follicular (differentiated) types followed by medullary carcinoma (in subjects harboring *RET* pathogenic variants, either germline in 20% of cases or sporadic in 80% of cases) and very rarely anaplastic/poorly differentiated forms. Papillary cancer represents 70–75% of all thyroid malignancies in non-endemic areas, while the follicular type accounts for 10–15% in non-endemic regions (respectively 30–40% in endemic areas); medullary thyroid cancer involves less than 5% of the thyroid malignancies. Other uncommon thyroid neoplasias include primary thyroid lymphoma, teratoma or squamous cell carcinoma [1,2,3]. While a great body of multidisciplinary evidence involves the topic of thyroid cancer, exceptional findings such as thyroid cancer in ectopic thyroid tissue (representing 0.3–0.5% of all thyroid cancers) require a complex panel of investigations in order to differentiate it from benign tissue and to decide the best surgical approach for an overall better outcome. Awareness remains the key operative element since the clinical index of suspicion is rather low, especially in non-cervical areas. Hence, currently, ectopic thyroid tissue remains a matter of individualized management [4,5,6].

### The Issue of Ectopic Thyroid Tissue

Generally, ectopic thyroid tissue represents an unusual finding that associates multiple challenges and pitfalls starting with its initial recognition. The prevalence is one to three cases per 100,000 (or one 1 to 300,000) people in the general population [7,8,9]. However, the true prevalence might be underestimated. Ectopic neck thyroid is part of the neck congenital masses (also including thyroglossal duct remnants, epidermoid/dermoid cysts, laryngocele, etc.) that are identified in 21–45% of the children presenting any type of neck mass, respectively, in 5% to 14% of the adults [10]. Among the population subgroup diagnosed with any type of thyroid disease, the prevalence of ectopic thyroid tissue stands for one case in 4000 to 8000 subjects [9]. Moreover, the prevalence of ectopic thyroid tissue regardless of the site in autopsy-based studies is 7% to 10% [11].

Ectopic thyroid tissue represents an exceptional developmental anomaly regarding the impairment of the gland movement from the primitive foregut to the pre-tracheal position. The gland originates from the endodermal diverticulum at the level of the first and second pharyngeal pouch; it descends to its normal cervical site (below the larynx and hyoid bone). The eutopic thyroid is situated anterior to the second, third and fourth tracheal rings. Hence, the ectopic thyroid tissue is situated anywhere between the foramen cecum (which is situated between anterior two thirds and posterior one third of an adult tongue) and mediastinal space (across thyroglossal tract). From the embryogenesis perspective, thyroid gland represents the first endocrine tissue that is developed during intra-fetal life. The thyroid primordium starts its development since weeks three to four of gestation, while its migration takes place between the weeks five and seven; the secretion of fetal thyroid hormones begins between weeks ten and twelve [12,13]. The transcription factors that play a role in thyroid development are TTF1 (thyroid transcription factor-1 or NKX2.1) and TTF2 (or Foxe1), PAX8 (paired-box gene), HHEX (hematopoietically expressed homeobox protein) and TSH (thyroid-stimulating hormone) receptor, and they may be involved in the ectopic presentation [7,11,14,15,16].

The most common ectopic thyroid site is represented by the tongue (the lingual thyroid is involved in 90% of all ectopic cases; of note, the first ectopic thyroid tissue at this level was described in 1869 by Dr. Hickman in a newborn with rapidly fatal outcome due to respiratory obstruction) followed by various locations (that are called sublingual type [17]) such as submandibular [18], peri-tracheal, larynx, sub-diaphragmatic area [12], etc. Across these 10% of ectopic cases, the most uncommon sites that are only partially understood based on the embryogenetic perspective are at the gastrointestinal level, including the gallbladder [19], adrenal glands, ovaries (struma ovarii), lumbar/renal [19,20,21,22], axillary [23], mammary [24], supra-sellar and suprachiasmatic [25]. The mediastinal location may be easily explained by the local attachment of the primordial thyroid at this site before starting its caudal migration [8,26]. Alternatively, ectopic thyroid tissue at the level of the anterior mediastinum, lung [27], heart, and pericardium may have been dragged into the chest together with the heart and its great vessels amid physiological embryogenesis [11,28,29]. Of note, carcinoma showing thymus-like differentiation (CASTLE) of the thyroid might also be found in the mediastinum in addition to EMTs and retrosternal goiter [30].

In ectopic thyroid tissues, a female to male ratio was found of three to four; the detection may be at any age; some authors appreciated that an early identification may be registered during the teenage years (even more common than in adults) [13]. A potential explanation for the more frequent detection in women is related to the sex differences with respect to the necessary thyroid hormones across puberty, menstruation, and pregnancy [7].

Almost half of the patients with ectopic thyroid are asymptomatic [11]. Some hypotheses suggested that ectopic mass might grow and become symptomatic in subjects experiencing a TSH increase of an unrelated cause (for instance, in trauma, infections or amid traditional primary causes of hypothyroidism, including congenital forms). Since TSH controls both the gland volume increase and the thyroid hormones secretion/production, the ectopic tissue might become visible under these circumstances. Yet, the current level of understanding of TSH control over the ectopic tissue and the balance between thyroid hormone production within the follicular cells from eutopic and ectopic tissue is still low [31].

Apart from specific endocrine considerations, the anatomical perspective includes two types of ectopic thyroid: aberrant (no eutopic thyroid) and accessory (associated with normal cervical thyroid gland), which is found in 75% of cases [7]. Rarely, multiple ectopic thyroid sites (dual or triple ectopic thyroids) are described within the same patient [7,32]. One third of children with ectopic thyroid tissue might experience hypothyroidism, especially the cases with aberrant patterns [31,33]. Ectopic tissue might have a lower rate of iodine uptake and thyroid hormones production than orthotopic glands; thus, cases with aberrant thyroid are associated with hypothyroidism more often (but ectopic tissue with hyper-function has been reported, too), while decreased radioiodine uptake might prove to be a source of bias in the adequate recognition of ectopic tissue amid imaging exploration [7,34]. All types of thyroid cancer have been reported in ectopic thyroid tissue as well as different areas of thyroiditis involvement [7,34,35,36]. Notably, some authors consider that the rate of malignant transformation in ectopic sites is similar with eutopic thyroid (particularly, for the papillary thyroid cancer [37]), but the level of statistical evidence in this particular matter remains low [8,38]. For instance, the 2023 study of Gao et al. [7] (which is described by the authors as being “the world’s largest single-center sample size of comprehensive ectopic thyroid gland diagnosis and treatment” that included patients treated between 2013 and 2022) investigated 47 patients (five subjects had a double ectopic tissue, thus a total number of 52 ectopic thyroids) with different ectopic presentations showed that 61.7% of them had an accessory thyroid tissue; 78.7% were females; average age at diagnosis was of 36 years (range between 4 months and 65 years); the average maximum diameter of the ectopic tissue was of 3.2 cm; the most common sites were lingual (N = 20/52 thyroid tissues), followed by submandibular (N = 10/52), latero-cervical (N = 10/52), mediastinal (N = 4/52), ovarian (N = 7/52), and esophageal (N = 1/52). A median follow-up of 59.4 months (range between 12 and 117 months) was described. The rate of malignancy was of 1/52 (this was a single case of papillary thyroid cancer at lingual tissue requiring its surgical removal followed by lifelong TSH suppressive therapy) [7].

We aimed to analyze management in cases with an ectopic mediastinal thyroid (EMT). Multiple insights are provided with respect to EMT-related thyroid cancer and non-malignant findings regarding the pathological (histological and cytological) report, the clinical presentation, the imaging traits on first identification and the secondary assessment (meaning the need for an additional imaging tool for diagnosis), the endocrine (hormonal) profile, the presence of connective tissue to the cervical (eutopic thyroid gland), the use of biopsy or fine needle aspiration (FNA), the choice of surgical techniques and post-operatory outcome. This was a comprehensive review based on revising any type of freely PubMed-accessible English, full-length papers with the keywords “ectopic thyroid” and “mediastinum” from inception until March 2024 (a total of 318 articles have been manually screened and relevant data were analyzed). We included original articles (case reports, series or studies) that specified data with respect to EMTs (Figure 1).

## 2. Ectopic Mediastinal Thyroid (EMT)

According to our methods, the earliest papers featuring EMTs were published in 1958 (aberrant thyroid as differential diagnosis for different mediastinal neoplasia) [39], in 1964 (the first specific mention of EMT analysis in terms of diagnosis and therapy) [40], and in 1983 (regarding the use of FNA in EMTs in the upper mediastinum) [41]. Generally, the EMT represents less than 1% of all mediastinal tumors (of any type) regardless of their origin and approximately 1% of all ectopic thyroid sites [42]. For instance, reports of thymomas, lymphomas, and germ cell tumors are more frequent in the mediastinum; alternatively, other findings are fibromas, lipomas, hemangiomas, cysts, and teratomas [29,42,43,44,45] as well as Castleman’s disease and metastases from different originating cancers [46]. Other uncommon conditions also associated with an distinct endocrine profile are represented by the intra-thoracic ectopic parathyroid glands [47] and paragangliomas [9,13].

### 2.1. Sample-Focused Analysis

We identified ten studies or case series (of at least three patients per series) that included the evaluation of EMTs among other outcomes (this is distinct for single case reports specifically describing patients with EMTs) [7,9,45,48,49,50,51,52,53,54] (Table 1).

To summarize, these are all retrospective studies (n = 10; N = 36 subjects with EMTs, and one them had a papillary thyroid carcinoma in the EMT [52]) with various endpoints [7,9,45,48,49,50,51,52,53,54]:

Stanford database (N = 7 patients with benign EMTs) was analyzed across two distinct papers, but this was the same cohort [9,48];

A single-center study on surgical outcome (between 1991 and 2006) amid approaching cervico-mediastinal goiters (N = 97 individuals) identified 11 of them as having a “forgotten” goiter in the mediastinum (an alternative name for the EMT) [51];

A study of 3092 patients who underwent thyroidectomy (between 2000 and 2013) identified 28/3092 of them with ectopic thyroid tissue of any type; among this subgroup, 5 out of 28 had EMTs (female to male ratio of 3 to 2; mean age of 41 years) [52];

A single-center study of ectopic thyroid tissue (N = 47) of any type (between 2013 and 2022) identified 4 out of these 47 individuals with benign EMTs (female to male ratio of 4 to 1, mean age of 55.5 years; of note, one subject had a synchronous thymus lipoma) [7];

A case series on quality assurance protocol in endobronchial ultrasound-guided transbronchial needle aspiration (EBUS-TBNA) included three EMT cases (female to male ratio of 2 to 1; average age of 84.33 years) [49];

A case series of benign EMT (N = 3 subjects; female to male ratio of 2 to 1, mean age of 52.33 years) focused on using endobronchial ultrasound-guided fine needle aspiration (EBUS-FNA) [50];

An eight-patient series with huge mediastinal masses identified one benign EMT in a 44-year-old male [45];

A retrospective study in 665 patients who underwent thyroidectomy (between 2005 and 2012) identified one subject with EMT [53];

A study of 16 patients with benign ectopic aberrant thyroid identified one patient with EMT [54].

### 2.2. Clinical Presentation and Scenario of Detection in EMTs

The presentation may be asymptomatic; also, there is the typical scenario of detection according to an incidentaloma for unrelated medical and surgical issues. Alternatively, local compressive symptoms such as dysphagia, dysphonia, dyspnea, cough, chest pain, stridor, sensation of retrosternal mass, Horner’s syndrome, mediastinal syndrome [13], and superior cave vein syndrome [55] have been reported. For example, a retrospective database study of a high-volume academic surgical center, using “ectopic thyroid” as the search word, identified 202 cases of ectopic thyroid tissue (any site); EMTs were found in 7/202 subjects which firstly pointed out the rarity of the condition (hence, a rate of 3.46% amid other ectopic locations was confirmed) [48]. Initial presentation (N = 6/7 patients) included compressive symptoms or hyperthyroidism, respectively, but one out of the seven subjects with EMTs was identified with the mediastinal mass as an incidental finding amid computer tomography (CT) scan (thus, 14.28% of the EMTs may be detected as an incidentaloma) [48]. Moreover, the retrospective single-center study of Gao et al. [7] showed a rate of mediastinal involvement amid other ectopic thyroid sites of 4/47 adults (8.5% of all ectopic thyroids). The identification of an EMT was as follows: one case had cough and sputum and the other three patients were incidentally detected (asymptomatic presentation); thus, a much higher rate than Aziz’s study [48] was reported [7]. Remarkably, after EMT suspicion, a re-assessment amid anamnesis might highlight mild complains that at first seemed irrelevant such as intermittent dyspnea, chest pressure, night sweats or facial erythema [9,48]. Another interesting detection was made via assays for newly detected Graves’s disease in an adult male patient of 41 years. The case published by Agrawal et al. [34] in 2019 showed an accidental imaging detection of a suspected EMT (that was never biopsied or removed) at 99m-Tc (Technetium) sodium pertechnetate scintigraphy followed by the second imaging scan in terms of SPECT-CT (single-photon emission computed tomography); due to an increased level of thyroid hormones, only a non-contrast, low-dose CT scan was performed [34].

One of the key messages remains the fact that accidental EMT detection does not necessarily mean a completely asymptomatic patient in the matter of EMTs. In addition, we mention that thyroid incidentalomas (at the eutopic gland) represents the most common type of endocrine incidentaloma (despite the fact that the term “thyroid nodule” is more frequently used in daily practice) [56,57].

In additional to these clinical elements that are directly connected to the presence of the mediastinal lesion, highly aggressive malignancy in EMTs might be detected due to local or distant malignant spreading as the first step of further undergoing EMT identification. Here, we introduce the cases with thyroid cancer in EMTs (regardless of the scenario of detection) according to our methods [36,37,41,42,55,58,59,60,61,62,63,64,65,66,67,68,69,70,71,72] (Table 2).

One short note: 19 papers introduced a single case report per article [36,37,41,42,58,59,60,61,62,63,64,65,66,67,68,69,70,71,72] and another publication presented two patients [55], leading to a total of 21 patients with any type of malignancy in EMTs. Specifically, primary thyroid cancer was identified in 20 patients (plus, a case of a one newborn displayed an immature teratoma hosted by the EMT [68]). Demographic features (N = 19, mean age of 62.94 years) showed a female to male ratio of 9 to 10 (for one subject, these data were not available [41]); the female group (N = 9) had a mean age of 63.4 years (range: 36 to 95 years); the male group (N = 10) had an average age of 66 years (range: 34 to 90) [36,37,42,55,58,59,60,61,62,63,64,65,66,67,69,70,71,72].

To summarize, the histological types of the primary thyroid malignancies in 22 subjects (Table 1 and Table 2) confirmed with EMT-related cancer were as follows:

Papillary (the most common type; the patients had any form from microcarcinoma to severe metastatic disease); of note, one more case was introduced in prior studies-based analysis; hence, there were a total of eleven subjects [52];

Follicular variant of the papillary type (N = 2);

Hürthle-cell thyroid follicular malignancy (N = 1);

Poorly differentiated (N = 1);

Anaplastic (N = 2);

Medullary (N = 1);

Lymphoma (N = 2);

MALT (mucosa-associated lymphoid tissue) (N = 1) [36,37,41,42,55,58,59,60,61,62,63,64,65,66,67,69,70,71,72];

An additional case of immature teratoma does not belong to the specific category of primary thyroid cancer, but it was identified in ectopic tissue (EMT) [68].

### 2.3. Exploring the Thyroid Panel in Patients Confirmed with EMT

The endocrine profile in EMTs is essential before and after EMT removal (if any); the thyroid anomalies might share the same pathogenic traits as the EMT or they are incidental, but this is still a matter of debate [73]. Nevertheless, one subject with an EMT might have a normal endocrine status or not, and this is an important part of the overall management of EMTs. However, across our research, specific data on thyroid profile were not always available. For instance, in the study of Aziz et al. [48] (N = 7), only 4/7 had endocrine assessments, and all of them (4/4) had normal TSH [48]. The mentioned study of Gao et al. [7] (N = 4) showed that all patients with EMTs had a normal thyroid function [7].

The sample-focused analysis according to our methods identified another 37 EMT adults (22 females and 12 males; specific demographic data were not available for three cases; mean age was of 56.32 years, females subgroup: mean age of 56.04, range: 30–80 years; males subgroup: average age of 56.83, range: 31–74 years; across 35 articles featuring a single case study and two articles introducing each two subjects [74,75,76,77,78,79,80,81,82,83,84,85,86,87,88,89,90,91,92,93]), whereas benign EMTs were associated with a normal thyroid panel in terms of function, negative autoimmunity, and lack of nodules/cancer/goiter in eutopic (cervical) gland before and after EMT removal or identification if the EMT was not resected (of note, we also, included the cases whereas no specific thyroid data were provided, thus, it was presumably normal) [13,29,74,75,76,77,78,79,80,81,82,83,84,85,86,87,88,89,90,91,92,93,94,95,96,97,98,99,100,101,102,103,104,105,106,107] (Table 3).

Moreover, across our research, we identified a heterogeneous spectrum of thyroid anomalies in subjects confirmed with EMTs coming as a second (non-EMT) thyroid disease (affecting the cervical eutopic gland) or connected to the EMT profile (for instance, reciprocally influencing the overall thyroid function or the iodine uptake between the cervical and mediastinal thyroid tissue). Generally, one patient with an ectopic thyroid might have been admitted for prior thyroid conditions in terms of regular check-up or an unexpected change in previous thyroid status. The insights of the concurrent (orthotopic) thyroid diseases have been reported in relationship with any location of the ectopic follicular/colloidal tissue [108,109,110,111].

This large frame of interplay includes thyroid dysfunction (hyperthyroidism and hypothyroidism, either clinically manifested or mild/subclinical); a history of thyroid surgery at the orthotopic gland (for benign or malignant conditions), and the co-presence of the thyroid nodules or multinodular goiter in the cervical thyroid or different forms of thyroiditis (Hashimoto’s thyroiditis or Graves’ disease) that were confirmed by serum antibodies against thyroid or by histological report [112,113,114]. The crossroads between thyroid dysmorphogenesis and thyroid dysfunction makes no exception for EMTs. The case-focused analysis pinpointing the patients confirmed with benign EMTs that also have been affected by any type of prior, concurrent, and even early post-operatory (after EMT removal) thyroid disease are displayed in Table 4 [11,22,34,35,73,87,102,115,116,117,118,119,120,121,122,123,124,125,126,127,128,129,130].

The cohort of patients with benign EMTs and thyroid diseases included a total of 23 patients across 24 papers [11,22,34,35,73,87,102,115,116,117,118,119,120,121,122,123,124,125,126,127,128,129,130] that have been reported as follows: 20 single case reports per article; two cases were reported in addition to another case that has already been introduced in Table 3 (two case reports per paper [87,102]); two papers addressed the same EMT patient from different perspectives [127,128]. The female to male ratio was 17 to 3 (for three patients, the demographic data were not available). The average age at EMT diagnosis was of 52.1 (range: 19 to 67) years; female group: mean age of 52.76 (range: 19 to 72) years; male group: mean age of 48.33 (range: 41 to 62) years [11,22,34,35,73,87,102,115,116,117,118,119,120,121,122,123,124,125,126,127,128,129,130]. Overall, 60 patients had benign EMTs (Table 3 and Table 4, meaning the analysis based on case reports), and 38.33% of them were also affected by a second condition of the thyroid status, women being more prone.

To summarize, a total of 117 patients with any type of EMT were described across the papers we identified according to our methods:

36 subjects (and one of them with a malignant EMT [52]) were confirmed in studies with various endpoints (other than specifically evaluating the EMT population) or case series of at least three EMT patients per series [7,9,45,48,49,50,51,52,53,54];

21 subjects diagnosed with any type of malignancy in the EMT [36,37,41,42,55,58,59,60,61,62,63,64,65,66,67,68,69,70,71,72] plus the mentioned case above [52] (N = 22 persons with malignant EMTs);

37 subjects with benign EMTs and otherwise normal thyroid profile [13,29,74,75,76,77,78,79,80,81,82,83,84,85,86,87,88,89,90,91,92,93,94,95,96,97,98,99,100,101,102,103,104,105,106,107];

23 subjects with benign EMTs and a secondary thyroid condition of any type [11,22,34,35,73,87,102,115,116,117,118,119,120,121,122,123,124,125,126,127,128,129,130].

#### 2.3.1. Thyroid Dysfunction in Patients Confirmed with EMTs

All kinds of thyroid function anomalies have been reported in subjects confirmed with EMTs; sometimes, the dysfunction and the ectopic tissue share common pathogenic traits [52]. Previous data in aberrant thyroid showed a higher rate of congenital hypothyroidism and an increased prevalence in the pediatric population, particularly for ectopic cervical/lingual thyroid [131,132,133,134]. Yet, amidst our sample-based database, only one case was confirmed with congenital hypothyroidism out of the 117 subjects, suggesting that perhaps other non-EMT sites are involved in this connection [52]. Generally, various genetic and molecular studies addressed the issue of congenital hypothyroidism (for example, anomalies of *TSHR*, *SLC26A7*, *JAG1*, *DUOX2*, and *FOXE1* gene) linking the thyroid dyshormonogenesis to thyroid dysgenesis [135,136,137,138,139].

Our sample-based analysis (Table 1, Table 2, Table 3 and Table 4) showed hypothyroidism as the following:

Congenital type [52];

Iatrogenic hypothyroidism (treated or untreated with levothyroxine replacement) following previous total or partial thyroidectomy for multinodular goiter [50,102,122], benign single nodule [60], toxic Plummer’s nodule [124], thyroid cancer in orthotopic gland [35,42,123], thyroiditis plus benign nodules [118,120];

Primary hypothyroidism other than congenital or iatrogenic type [37,73].

Of note:

Mild (subclinical) hypothyroidism was identified in two reports [73,126];

Transitory hypothyroidism following EMT removal was found in a single case of 53-year-old female [73], suggesting the usefulness of thyroid assays following EMT resection.

Hyperthyroidism in patients who were confirmed with EMTs was identified on admission based on blood assays; however, a normal thyroid function might not exclude an increase iodine uptake in EMTs or in eutopic thyroids (with suppression of the other tissue); thus, the functional imagery might prove an important point in the matter of thyroid tissue exploration [140,141,142]. For instance, Kumaresan et al. [99] showed in a young lady an autonomous activity of the EMT according to 99m-technetium pertechnetate scintigraphy that suppressed the uptake of the eutopic tissue (yet, associating an overall normal thyroid function according to the blood assays on first admission); a recovery of normal thyroid gland function was registered when the scintigraphy was repeated six weeks following EMT removal [99]. The case reported in 2021 by Kola et al. [73] introduced a 42-year-old smoker male who was admitted for a 3-month history of dyspnea, chest pain, and fatigue. He was found with mildly low TSH (of 0.33, normal: 0.35–4.94 mU/L). CT showed a tumor at anterior mediastinum of 9 by 6 cm (with heterogeneous structure at CT scan). He also incidentally had a thyroid nodule on the left lobe of 0.9 cm and a right adrenal tumor of 5 by 3 cm, which was confirmed by MRI (magnetic resonance imagery). The 99m-Tc pertechnetate scintigraphy showed an orthotopic thyroid with a mildly reduced uptake rate (a potential prior asymptomatic thyroiditis that also induced TSH lowering was retrospectively suggested), but the chest scan was not included, since an EMT was not suspected at that point. Finally, the mediastinal mass was surgically removed. A post-operatory benign EMT was confirmed while TSH, T3 and T4 were found normal one month after surgery [73]. Serim et al. [121] introduced in 2016 a lady with a 2-year history of hyperthyroidism having a hyper-functional EMT but a normal thyroid (eutopic) gland according to 99m-Tc scintigraphy that actually highlighted the delicate aspect of navigating thyrotoxicosis blood serum confirmation via scintigraphy profile [121].

Hyperthyroidism/thyrotoxicosis status involved the following (according to our sample-focused analysis):


Clinically manifested [34,52,99,116,121,122] or subclinical [73,125] forms;

One case switched from levothyroxine replacement upon prior thyroidectomy to thyrotoxicosis that was persistent after hormone administration stopped and lead to the identification of TRAb (anti-TSH receptor antibodies) positive Graves’s disease in the EMT [122];

One case was under tapazol for hyperthyroidism at the moment of EMT identification [129].

#### 2.3.2. Elements of Thyroiditis in Orthotopic and Ectopic Tissue

Thyroiditis stands for the presence of autoimmune chronic Hashimoto’s thyroiditis caused by antibodies against thyroid, namely anti-thyroperoxidase antibodies (antiTPO) or anti-thyroglobulin antibodies (antiTg) usually causing hypothyroidism or, at the other end of the same autoimmune spectrum, Graves–Basedow’s disease (with or without thyroid eye disease) due to serum thyroid-activating antibodies TRAb [143,144,145]. The diagnosis itself may be established not only via high serum levels of the antibodies (which does not point out any difference between orthotopic or ectopic thyroid involvement), but, also, based on cytological and/or histological exploration, respectively, according to highly suggestive ultrasound features in the eutopic (cervical) gland. All these methods of thyroiditis confirmation have been reported in patients confirmed with different types of thyroid dysgenesis, including ectopic tissue [146,147,148,149].

To summarize, we found the following in the thyroiditis profiles of patients confirmed with EMTs:

Positive serum antibodies [63];

Positive serum antibodies and pathological confirmation of thyroiditis in eutopic gland [36,118,120];

Positive serum antiTPO and antiTg in addition to thyroiditis confirmation in both cervical thyroid and an EMT [59];

Histological report of EMT (but not in cervical eutopic thyroid) confirming focal lymphocytic thyroiditis in a 56-year-old male [105];

In the matter of TRAb positive status, the full-blown picture of Graves’s disease on first admission was registered in one male of 41 years that was finally identified to also have EMT [34]; another case, a 67-year old female who underwent total thyroidectomy 7 years before for non-toxic goiter, had a TRAb positive EMT [122] or previous history of Graves’s disease in one case (9 year before EMT diagnosis) [130];

Inflammatory thyroiditis pattern in benign EMTs (post-operatory confirmation) [102];

Retrospective diagnosis of thyroiditis in eutopic thyroid due to otherwise unexplained reduced uptake amid 99m-Tc pertechnetate scintigraphy [73].

#### 2.3.3. Thyroid Nodules/Multinodular Goiter and Cancer in Eutopic Thyroid or Prior Thyroidectomy before EMT Detection

Generally, thyroid nodules in cervical gland represent the most common endocrine finding affecting between 5% and 20% of the global population depending on the study, geographic region (iodine deficient areas) and methods of detection [150,151,152].

Nevertheless, EMT subjects might display certain elements:

As mentioned, some EMTs were detected amidst investigations following a history of thyroidectomy for nodules or the co-presence of a multinodular goiter, autoimmune thyroiditis or Graves’s disease [50,102,122].

The suspicion and consecutive confirmation of EMTs required the exploration of neck thyroid as a mandatory evaluation and, in cases with thyroid nodules in cervical gland, thyroid FNA was performed [11,22,58,59,69,80,87,98,117].

Moreover, upon differentiated thyroid cancer confirmation in EMTs, total thyroidectomy was mandatory in order to start radioiodine ablative therapy (if thyroidectomy was not synchronous with the sternotomy or previously completed for unrelated issues) [55,58,59,60,61,62,64,69,71].

Also, some cases of EMT removal, particularly, the masses located within the upper mediastinum, required thyroid resection for a better access in order to avoid a sternotomy [88,94,96].

Concurrent multinodular goiter was described in 4/5 patients in Santangelo’s study [52] and across other single case reports [117,125,127] or single simultaneous thyroid nodules [22,54,73,87,115] or thyroid cancer in eutopic thyroid [11,22,42].

#### 2.3.4. Serum Tumor Markers in Individuals Confirmed with EMTs

Serum tumor marker thyroglobulin is useful as a prognostic marker for differentiated thyroid cancer after its removal [153,154,155]. Its utility in benign conditions is limited unless the presence of ectopic tissue and some authors considered its very high value of being a predictor for an ectopic thyroid [55]. Of note, exceptionally elevated levels are found in malignant EMTs with metastatic disease [58].

Looking at the thyroglobulin assays in EMT patients, we conclude the following:

High values might suggest a synchronous ectopic tissue and intact eutopic thyroid gland, but the index of suspicion remains low;

High values after thyroidectomy of eutopic gland might suggest an ectopic tissue with an increased index of EMT suspicion;

High values after thyroidectomy with post-operatory differentiated thyroid cancer confirmation should differentiate ectopic tissue (including EMTs) from local remnants, recurrent or metastatic thyroid malignancy;

High values after surgery for differentiated thyroid cancer in eutopic and ectopic thyroids suggest a disease relapse or spreading, thus indicating the need for a whole body iodine scintigraphy (if confirmed, further additional surgery or a new dose of radioiodine ablative therapy should be included in the overall patient management) [36,37,41,42,55,58,59,60,61,62,63,64,65,66,67,68,69,70,71,72,122].

#### 2.3.5. Other Blood Assays in EMTs: Endocrine and Non-Endocrine Elements

Further blood assessments amidst EMT confirmation may include the following:

Investigations serving for differential diagnosis of the mediastinal mass such as an ectopic parathyroid tumor causing a primary hyperparathyroidism (blood calcemic and PTH assays) [54,81,115] or a paraganglioma (requiring measurements of blood and 24 h urinary metanephrines and normetanephrines at least twice) [9,76,81,102,105]. Of note, an ectopic (mediastinal) parathyroid tumor is more frequent than an EMT [156,157].

Calcitonin and carcinoembryonic antigen are mandatory for the diagnosis and surveillance of the medullary thyroid carcinoma [65].

Baseline ACTH (adrenocorticotropic hormone) and cortisol assays (baseline and dexamethasone suppression test) are necessary for the confirmation of an ectopic ACTH syndrome [65]. Notably, there is only an EMT case (N = 1/117) affected by this complex condition, but this paraneoplastic syndrome has been reported in relationship with the eutopic gland-related medullary malignancy or in non-endocrine cancers such as primary pulmonary carcinomas [158,159,160,161].

The biochemistry panel might show elevated serum total calcium in primary hyperparathyroidism or hypopotassemia in Cushing’s paraneoplastic syndromes [65].

Hemogram revealed an increased number of eosinophils and white blood cells (leukemoid reaction) in a fulminant cancer (e.g., anaplastic) in the absence of concurrent infections [66].

Synchronous (suspected or confirmed) infections as part of the scenario concerning EMT detection such as lung actinomycosis [78] or, recently, COVID-19 [76] have been reported.

One single case associated an adrenal incidentaloma [73]; thus, the entire panel of adrenal hormones that serve as screening tests should be considered if incidental adrenal masses are detected at CT, MRI or PET-CT scans [162,163].

### 2.4. Imaging Features in EMTs

#### 2.4.1. CT Scan

CT scan represented the most important and the mostly used imagery evaluation, and it was a mandatory step of approaching (suspected or confirmed) EMTs [7,11,13,29,35,37,42,48,55,58,59,60,61,62,63,70,71,73,74,75,76,79,80,82,89,92,93,95,96,100,102,103,105,106,117,118,119,120,121,122,124,125,129]. The largest diameter varied from 1 to 15 cm; no correlation between the size and a malignancy trait in EMTs could be established. The size groups (according to the largest diameter via our practical perspective) may be regarded as follows: <2 cm [22,59,81,89,118]; between ≥2 cm and <3 cm [11,35,55,83]; between >8 cm and <10 cm [13,73,102,117]; more than 12 cm [37,45,60,115]. The most common groups were as follows: between ≥3 cm and <5 cm [9,58,62,80,85,86,93,95,103,106,119]; between ≥5 cm and <6 cm [29,74,78,79,98,100,121,122]; between ≥6 cm and <8 cm [50,55,64,70,75,82,84,88,92,100,120,123,124].

Calcifications [13,76,115,117,121,123], heterogeneous features [80,85,86,87,95,115,121], and even necrosis [13] in EMTs were described at CT scan. They do not seem to be associated with a malignant pattern in EMTs. This is contrast to the clinical significance of identifying micro-calcifications in papillary thyroid carcinomas at the cervical gland (underlying psammoma bodies) [164,165,166,167]. In EMTs, these traits highlight a long-standing condition or the potential of a further enlargement caused by a local necrosis/hemorrhage rather than a thyroid malignancy in EMTs, but further studies are necessary.

The upper mediastinal site was the most frequent; usually, EMTs were in the anterior mediastinum, and only a few were posterior [7,22,48,77,103,124,126]. For instance, in the study published in 2023 by Aziz et al. [48], all patients (N = 7) had an initial CT scan (a description of the location is useful by using the model of the four compartments: upper (superior) compartment (N = 3/7); middle compartment (N = 1/7); anterior compartment (N = 1/7); posterior compartment (N = 2/7), thus suggesting that the upper area is the most frequent. Also, apart from CT scan, further imaging work-up seemed useful (N = 3/7): one subject had a radioiodine-based scan; another one underwent a PET scan and another was explored via cervical MRI [48]. Gao’s [7] study showed these 4/4 EMTs were within the upper mediastinum (including one posterior to trachea and one within the thymus area). The maximum diameter was larger than 2 cm in all four cases; all were accessory type, not aberrant. Radioiodine scintigraphy was not used in any case, while CT was performed in all cases [7]. The use of SPECT-CT with 99m-Tc sestamibi may confirm ectopic parathyroid tumors at the mediastinum level [168].

Intra-cardiac/intra-pericardial EMTs represent a challenging situation; however, an adequate intervention might provide a good outcome [29,79,82,83,86,89,90,119]. Kocaman et al. [82] reported in 2020 an intra-pericardial EMT, while in 2018, an adult case of EMT near the left atrium in a renal cancer survivor was introduced [86]. Another presentation of a female adult showed a successful removal of a heart-located EMT (right ventricle) [89]. Similarly, a 63-year-old woman was accidentally detected with a nodular intra-pericardial lesion while having a normal thyroid function. CT showed a heterogeneous structure; coronary CT angiography revealed no connection to the coronary artery; (^18^F-fluorodeoxyglucose) ^18^F-FDG-PET was added to further exploration that concluded with a trans-sternal approach. Post-operatory EMT confirmation (2.5 by 2 by 1.5 cm) was followed for one year and showed a good outcome [83]. This paper of Sato et al. [83] was published in 2019, and the authors showcased that since its first description in 1986, six intra-pericardial cases (included this one) were published with a median age of 49 years. In this intra-pericardial instance, the main focus points were the blood supply and the relationship with the coronary artery as being essential amid the entire case management and the fact that, unless surgically removed, thyroid tissue confirmation is most likely not feasible in this particular location despite advanced imagery scans [83].

As mentioned by Gao et al. [7], a double ectopic thyroid has been rarely reported in EMT cases (less often than lingual sites) [7,22,75,93]. Nagireddy et al. [75] introduced a 57-year-old lady with double EMTs. The presentation included a one-month history of cough and chest discomfort. CT was provided and showed two masses of 7 by 7 cm and 4.9 by 5 cm (they were on the both sides of the upper mediastinum, on the right and the left, respectively). A CT-guided biopsy-based pathological report revealed a colloidal goiter (at right tumor mass). Due to the vascular proximity, sternotomy was completed with both tumors excision [75]. Wang et al. [93] presented a 45-year-old female with a double ectopic thyroid; one was an EMT and the other one was a lateral cervical tissue (according to the authors of this paper published in 2014, this was the first case with a double ectopic thyroid in this combination: neck and mediastinum). An EMT was removed via VATS (video-assisted thoracic surgery) in addition to using a neck incision. Both tissues were completely separated from the eutopic gland and resected [93].

#### 2.4.2. The Use of ^123^ or ^131^Iodine or 99m-Tc Thyroid Scintigraphy

The second most important imagery tool was represented by the functional imaging of the thyroid tissue. If an EMT was suspected, this tissue was targeted using whole-body (not just cervical) scintigraphy; tracers such as 99m-Tc, ^131^ or ^123^iodine play a pivotal role despite variations of radioiodine uptake in ectopic tissues [6,42]. Sometimes, the metabolic rate in ectopic thyroid tissue is different from the eutopic gland, and thus the diagnosis might be missed [13,169,170]. Also, autonomous activity (hyper-function) in one thyroid tissue might suppress the uptake on the other tissue and, essentially, an EMT might not uptake 99m-Tc or the iodine tracer at all, thus being a source of potential bias. Iodine scintigraphy was used in the benign EMT/eutopic thyroid gland, and it is mandatory across the overall management of differentiated thyroid cancer [36,52,53,65,71,76,101,123]. Alternatively, a normal uptake in one thyroid lobe in addition to the lack of the other lobe (that was prior removed) and a reduced iodine uptake in the EMT was described in one case [124]. 99m-Tc scintigraphy was equally used for EMT diagnosis [34,69,71,73,87,91,93,102,106,115]. Various aspects should be kept in mind: for example, suppression of the eutopic gland due to EMT hyperactivity, and six weeks after EMT removal, a recovery of the normal uptake in cervical gland was registered [99] or an increased Tc uptake in EMTs and normal thyroid capture was described, while the overall hormonal assays were consistent with thyrotoxicosis diagnosis [121].

#### 2.4.3. Other Imagery Assessments

Second-line imagery tools following CT scan (and distinct from iodine/99m-Tc scintigraphy) included a large scale of investigations across our search. MRI was used for evaluating the cervical gland or distant metastases [22,53,58,59,99,100,102]. Alternatively, for bone metastases, whole body bone scintigraphy was applied [60,92]. PET-CT was performed for a prior diagnosed malignancy, for (suspected or confirmed) EMT cancer or related metastasis [37,55,58,59,61,84] showing hypermetabolic lesions. Benign EMTs might not uptake the tracer (and this stands for a potential pitfall in daily practice) [85,89,91,101], or it might actually reveal the tissue [11,48,83,87]. Rajaraman et al. [116] addressed the role of SPECT in highlighting eutopic and heterotopic thyroid glands, including patients with active thyrotoxicosis (hyperthyroidism otherwise is a contra-indication of iodine administration amid using intravenous contrasts for CT scans) [116]. SPECT-CT might be used to differentiate EMTs from ectopic parathyroid tumors [34,81,86,116].

As mentioned, the misinterpretation of various imaging procedures (that were actually performed for unrelated conditions) might lead to EMT detection. Chen Cardenas et al. [81] reported two patients who were misdiagnosed as paragangliomas during iodine-123/iodine-131 (^123^I/^131^I)-metaiodobenzylguanidine (MIBG) scan without an adequate iodine blockade at the eutopic gland. After mediastinoscopy-based resection, an EMT was confirmed by a post-operatory histological report [81]. An MIBG scan should be administered if suspicious of mediastinal paragangliomas [59,86]. Of course, ectopic thyroids at the neck level require the standard (and routinely performed) ultrasound evaluation that is not feasible for the mediastinal site unless at a very high (upper) EMT position [8].

To conclude, three main features of the ectopic thyroid, particularly the EMT, should be kept in mind: distinct blood vessels supply are found in ectopic versus eutopic tissue; another aspect is the fact that despite the presence of an EMT, there is a distinct orthotopic thyroid gland in terms of anatomy, hormonal profile, and histological report with an additional interplay (with concern to functional imagery), while a different pathologic profile/metabolic rate/tracer uptake in ectopic versus eutopic thyroid was reported in some cases.

### 2.5. Analyzing the Co-presence of the Eutopic (Cervical) Thyroid Gland

Ectopic tissue is identified in the presence of normal thyroid or in the absence of a eutopic (orthotopic) gland, which might represent a milestone with regard to the surgical procedure (the need to remove the cervical thyroid in order to access the EMT), or the identification of a differentiated thyroid cancer in EMTs requires further radioiodine therapy; thus, the cervical thyroid should be removed before applying it. Moreover, the virtual risk of iatrogenic hypothyroidism in subjects who lack the cervical thyroid or those to whom thyroidectomy was already administered has been already mentioned. Yet, according to our EMT-based analysis, the presence of a synchronous normal eutopic thyroid means that the EMT removal will most likely not cause hypothyroidism. On a larger scale, ectopic intra-thoracic thyroid tissue includes EMTs, but mostly substernal (or retrosternal) goiter (representing the actual extension of the cervical thyroid enlargement to the mediastinum, also involving the extension of an aggressive thyroid cancer beyond the thyroid bed [171,172] in addition to the metastases from a malignancy originating from the eutopic thyroid (including pre-sternal metastasis have been reported, as well [173]).

Most retrosternal goiters are developed within the upper and middle mediastinum and only rarely at the level of posterior compartment. Retrosternal goiter may be asymptomatic or might cause compressive symptoms according to local anatomical elements (such as dyspnea, dysphagia, cough, stridor, hemoptysis, etc.). Retrosternal goiter should be removed because of local effects (including acute respiratory insufficiency, upper cave vein syndrome, aspiration pneumonia, etc.), risk of expansion due to further enlargement (grow) and hemorrhage, respectively, risk of gaining toxic activity (hyper-function causing hyperthyroidism), and risk of malignant transformation (which is rather uncommon). The removal, depending on the location, size and surgeon’s skills, is either performed trans-cervical or via sternotomy or through thoracotomy [74,174]). In contrast to the retrosternal goiter (that has no separation line from the orthotopic thyroid and shares the same vessels), an EMT has its own blood supply (intra-thoracic vessels) [42]). The distinction between retrosternal goiter as an extension to an enlarged cervical gland and an EMT as a standalone tissue with encapsulated appearance can elegantly be made intra-operatory and/or after a post-surgery histological report in exceptional instances as reported in two cases by Sohail et al. [117] in 2019, but otherwise, the pre-operatory imagery exploration amid CT use and iodine-based scintigraphy should clarify the issue of retrosternal goiter versus an EMT [117].

### 2.6. Connective Tissue between EMT and Cervical (Eutopic) Thyroid

Most EMTs do not showcase a connective tissue to the cervical (physiological) thyroid (and this aspect is confirmed by our sample-based analysis), an EMT being distinct from a substernal goiter [13,48]. A thin strip of tissue between one thyroid lobe and an EMT was shown in a single case at contrast-enhanced CT [88]. Some authors agreed than only 2% (between 1% and 5%) of the intra-thoracic thyroid tissue is represented by an EMT, and the rest mostly involves a substernal goiter, which otherwise is very rare when compared to the traditional cervical goiter that comes as an easy-to-perform diagnosis due to routine neck ultrasound [48,83]. Hence, data on connective tissue are mandatory for this differential diagnosis. They may be provided based on imaging scans, intra-operatory identification or after surgery according to the histological report. Some authors suggested that potential common pathogenic factors are involved in these entities, substernal goiters and EMTs [175].

### 2.7. Pathological Report in EMTs

Firstly, there is the recognition of the ectopic thyroid remnants since the clinical index of suspicion is decreased followed by the biopsy/FNA and/or surgical removal depending on the EMT site, size, vascularization, and co-morbidities; overall, a decision of a multidisciplinary team is mandatory. Upon pathological confirmation of a thyroid cancer in most ectopic tissues, total thyroidectomy of the eutopic gland is necessary followed by radioiodine ablation and long-term TSH suppression therapy. Whether neck lymph nodes dissection is part of the second surgical step is a matter of personalized approach; for instance, it depends whether the ectopic gland was found in the neck area, if imaging evaluation suspected a lymph node metastasis, etc. [8].

While some studies in EMT subjects found no EMT malignancy [7,48], cancer (of any histological type) was found in 22 subjects out of the 117 (representing 18.8% of the entire cohort), according to our methods [36,37,41,42,52,55,58,59,60,61,62,63,64,65,66,67,68,69,70,71,72], which is an unexpectedly high rate (Table 1 and Table 2, Figure 2).

#### 2.7.1. Differentiated Thyroid Carcinoma

The earliest reports of malignant EMT were provided in 1983 (follicular type) [41] and 2003 (papillary type, specifically, columnar cell carcinoma) [71] according to our PubMed search. Differentiated type was the most frequent (N = 14/22 subjects with malignant EMT), papillary being far more frequent than follicular (N = 13/14 patients with differentiated malignancies in EMT) [36,37,41,42,55,58,59,60,61,62,63,64,65,66,67,69,70,71,72] (Figure 3).

#### 2.7.2. Anaplastic/Poorly Differentiated Thyroid Carcinoma

Yet, according to our research, a fulminant evolution was reported by Camagos et al. [66]. They published in 2010 a first case of a 95-year-old woman who died of severe respiratory insufficiency due to an EMT and associated anaplastic carcinoma but normal eutopic gland (as shown by the necropsy) [66]. In addition, Nguyen et al. [37] reported in 2023 the case of a 90-year-old male who suffered from prior non-surgical hypothyroidism and survived a prostate cancer two decades prior. He was admitted for a rash at the pre-sternal level in addition to a recent massive weight loss. A CT showed a tumor from manubrium to the mid-sternal body of 12 by 11 cm. A skin biopsy was performed and showed a papillary part (that was positive for AE1/AE3, TTF1, and PAX8) and a poorly differentiated component (that was CD58 positive in giant cells). This was followed by excisional biopsy of the mediastinal tumor. A confirmation of a rare anaplastic type was re-done, namely, the giant-cell-rich type associated with skin and sternal metastases. PET-CT identified lymph nodes metastases at the cervical and axillary level. The patient tested positive for *BRAF V600E* and survived 6 months since the first admission. Notably, the cervical thyroid was atrophic [37]. Median survival in anaplastic thyroid carcinoma at the eutopic gland is of 3 to 6 months. Generally, pathogenic variants of *TERT, TP53, BRAF*, and *RAS* genes have been reported in this malignancy. The same genetic pressor seems to act in ectopic malignancy as well. As seen in the eutopic site, the level of statistical evidence remains very low [37,176,177,178].

To summarize, the thyroid malignancies originating from the follicular cells at the cervical thyroid or EMT (meaning papillary, follicular, poorly differentiated and anaplastic types) embraced various patterns in EMTs as following: subject affected by a primary thyroid cancer in the eutopic gland but benign EMT [22,35,123]; multifocal carcinoma affecting both EMT and a eutopic thyroid [36,42,55]; one case was considered to have metastasis in the thyroid from an EMT cancer [55]; malignant EMT with benign features in the eutopic gland [37,41,52,55,58,60,61,62,64,66,69,70,71,72]. Notably, the distant metastatic spreading from an EMT was identified in bone [37,55,58,60,61] and pre-sternal skin [60]. Intra-mediastinal ectopic tissue was accidentally detected during surveillance or check-up for non-thyroid (unrelated) malignancies, and we identified seven such cases (a rate of 5.98% amid all EMTs); the primary carcinomas were located in the pulmonary [11,49,101], mammary [29,92,125] and ovarian [89] areas (Appendix A).

#### 2.7.3. Thyroid Lymphoma in EMT

A single case of mucosa-associated lymphoid tissue (MALT) lymphoma in an EMT was identified (in 2020). This was a 67-year-old lady who was accidentally detected at CT scan with an upper mediastinal mass of 2 by 1.3 cm situated anterior to the trachea (poorly enhanced structure) without connection to the cervical thyroid. Further MRI was completed as well as ^131^iodine MIBG scintigraphy that excluded a paraganglioma. A resection via transverse neck incision was provided and confirmed an MALT lymphoma and chronic thyroiditis. The mass was 1.5 by 1.2 by 0.9 cm with lymphoma cell infiltration to follicular epithelial cells (which were positive for thyroglobulin); B lymphocytes were positive for CD79a. An FNA of the ortothopic thyroid showed thyroiditis (no MALT). Post-operatory thyroid function remained normal. Further ^18^F-FDG PET-CT showed a thyroid accumulation. No other therapy was added; she continued surveillance [59]. Wu et al. [63] introduced a mostly challenging case of an adult who survived renal transplant 11 years prior; he developed a neck swelling and an upper mediastinal tumor that turned out to be a primary mediastinal large B-cell lymphoma. The immunosuppressive therapy for more than a decade amid being a renal recipient in a patient who three years before was confirmed with an autoimmune Hashimoto’s thyroiditis might imply a pathogenic contribution to this type of lymphoma yet with an exceptional location in EMTs [63]. Primary ectopic thyroid B cell lymphoma arising from an ectopic thyroid in the mediastinum was first reported in 2009 [67]. Of note, chronic thyroiditis represents the ground of the primary thyroid lymphoma; its incidence in patients with Hashimoto thyroiditis has been reported as 16 cases at 10,000 persons per year [59,179,180,181].

#### 2.7.4. Other Primary Malignancies in EMTs

As mentioned, a single case of its kind was published in 2011, namely, a mediastinal medullary thyroid carcinoma admitted for severe hypokalemia due to ectopic ACTH production [65]. Also, a dramatic pediatric case (the singular report in children we identified across our methods) involved an immature teratoma in EMTs [68].

### 2.8. EMT Biopsy and Fine Needle Aspiration Cytology (before or instead of Surgery)

Direct EMT access is provided by the biopsy such as transthoracic needle biopsy (TTNB) or CT-guided (or ultrasound-guided for EMTs located within superior mediastinum) FNA or EBUS-TBNA [13,182]. Generally, since most ectopic thyroid tissues are located in the neck areas, ultrasound-guided FNA-based cytology represents the most important pre-operatory tool for directly accessing the thyroid mass [7]. Gao et al. [7] had a confirmation of the thyroid tissue by FNA cytology (N = 3/4), and one case was resected via a thoracoscopic approach (surgery was performed only in this case; the others were conservatively followed upon providing the cytological analysis) [7]. FNA offers the cytological report, not the histological report, which for the orthotopic thyroid stands for the most useful tool to directly access the gland. One major pitfall concerns the follicular pattern in cytological testing, since similar features are found in thyroid adenomas and carcinomas [183,184,185]. As a general note, FNA remains a most useful tool for an ectopic thyroid within the neck area while for EMTs, a trans-esophageal or trans-bronchial approach is frequently required [49,76,186]. Metastatic thyroid carcinoma should be differentiated from EMTs (this is one of the reasons for meticulously checking the eutopic gland including thyroid nodules exploration via FNA once an EMT is suspected) [31,73].

Notably, Vuorisalo et al. [49] showed a three-case series according to a quality assurance program, whereas EBUS-TBFNA was administered in mediastinal lymph nodes in order to differentiate EMTs from metastases originating from follicular (cervical) thyroid cancer. An open issue might be represented by the high inter- and intra-laboratory variability in this particular matter [49]. Similarly, Tsai et al. [11] also raised the issue of mediastinal lymph nodes involvement in a relationship with non-thyroid cancers (such as those originating from lung) that should be differentiated from EMTs, especially if the subject is already known to have a history of a specific malignancy. This was a 50-year-old lady with a previous diagnosis of pulmonary adenocarcinoma (at the left upper lobe) who underwent surgical resection (segmentectomy) one year prior. At some point, she experienced headache, nausea and dizziness; thus, she came in for a complex imaging check-up, and two brain metastases were detected at MRI. She was offered gamma knife therapy. Moreover, CT and PET-CT scans for follow-up showed an enlargement of lymph nodes within the mediastinum and thyroid nodules. FNA at the thyroid level showed a papillary thyroid cancer. EBUS-TBFNA of the lymph nodes revealed they were cancer-free, as they were confirmed as being an EMT (2.9 by 1.6 cm). Of note, these mediastinal lesions at PET-CT were not hypermetabolic; thus, an alternative diagnosis to a cancer spreading was already taken into consideration before FNA results [11]. Notably, an immunohistochemistry panel in lung adenocarcinoma also reveals positive Ck7, TTF1, as seen in EMTs or metastatic papillary thyroid carcinoma, while PAX8 can only be found positive in thyroid tissue, as well as thyroglobulin stain; on the other hand, CD56 is positive in thyroid tissue and negative in pulmonary adenocarcinoma, but it may be positive in pulmonary typical and atypical carcinoids (whereas the well-known panel of blood neuroendocrine markers should be checked in terms of chromogranin, neuron-specific enolase, and synaptophysin; in this specific instance, TTF1 is inconstantly positive) [11].

Alternatively to FNA or EBUS-TBFNA, TTNB was used for histological rather than cytological analysis; for example, Sadidi et al. [74] performed TTNB to access an EMT located at the upper-middle mediastinum in a 54-year-old female who was admitted for cough and dyspnea for 3 years. The pathological report upon biopsy showed an adenomatous goiter; the procedure was followed by EMT removal via thoracotomy with a good post-surgical outcome. The largest diameter was 7.5 cm [74].

Overall, direct access to EMTs and/or the cervical (eutopic) thyroid was provided across a highly customized approach: mediastinoscopy for EMT diagnosis [9,87]; EBUS-guided biopsy [9,118]; trunk biopsy [13,60]; CT-guided core biopsy [69,75,92,115]; TTNB [74]; endoscopic ultrasound-guided biopsy [76]; CT-guided FNA in EMTs [96,99,101]; FNA/FNAC in EMTs with malignancy confirmation followed by its removal [7,41,58]; EBUS-FNA [50]; FNA in EMTs and cervical thyroid [80]; ultrasound-guided FNA in cervical thyroid [11,59,69,87,98,117]; EBUS-TBNA in EMTs [11,91] (of note, this investigation might be non-diagnostic [55,103] or might offer an adequate diagnosis, thus allowing the decision of surveillance, not EMT resection [62,89,95]). Some patients diagnosed with EMTs were directly referred for EMT removal after imaging evaluation, and no biopsy or cytology was completed pre-operatory. On the other hand, in small-sized EMTs, the confirmation of a benign EMT was not followed by a resection but rather by a conservative management under periodical check-up. Additionally, skin biopsy was completed in one case with pre-sternal spreading [37] or FNA in clavicle metastases [61].

Overall, the histological (or at least cytological) analysis is mandatory amid EMT exploration for a positive diagnosis, for identifying an EMT or for differential diagnosis with other malignant/benign mediastinal masses. The co-presence of the thyroid nodules in patients with EMTs requires at least FNA at this level. Non-diagnosis across different types of EMT biopsies or FNAs is not so rare; thus, an adequate EMT recognition may be affected. Also, in subjects with prior cancers, histological (or at least cytological) EMT testing is essential to differentiate it from metastasis. In patients with synchronous or prior thyroid nodular conditions (single nodules, multinodular goiter or cancer) in a eutopic gland, the second opinion (histological/cytological) should be made in relationship with the EMT profile (to check if they are similar). Metastatic lesions showing a follicular thyroid pattern may be originating from the EMT or cervical thyroid malignancies. An immunohistochemistry report (thyroglobulin, PAX8, TTF1, Ck7, and even lymphoma immune phenotyping) [55] adds value in confirming the thyroid profile in EMTs, especially in cases when pre-operatory or pre-biopsy investigations (particularly, iodine or 99m-Tc scintigraphy) were not consistent with a confirmation of ectopic thyroid tissue or in subjects suspected or confirmed with other malignancies [186,187,188,189,190].

### 2.9. Surgical Procedures to Remove EMTs

Median sternotomy or posterolateral thoracotomy represents the traditional approach of EMT, preferably via a minimally invasive approach as seen in recent years. Moreover, cervical incision (with immediate availability of sternotomy in case of intra-operatory complications) is preferred depending on the upper EMT location, surgeon’s experience, as well as eutopic thyroid proximity or prior/current thyroid removal (since performing a total thyroidectomy allows better access to EMT or retrosternal goiter resection to avoid a sternotomy upon skillful vessels ligature before removal) [9,13,94,96,187,191]. The indication of sternotomy depends on the EMT location and volume, vessels anatomy, the risk of hemorrhage and the pre-operatory identification of mediastinal lymph nodes enlargement without specified significance (potentially malign spreading) and the suspicion of a malignant tumor [75,192,193]. Also, as noted in the case of double upper mediastinal EMTs, synchronous removal of both tumors also required sternotomy [75].

In 2020, Imai et al. [80] reported a trans-cervical EMT resection without orthotopic thyroid removal that was feasible due to the very high position of EMTs within the upper mediastinum (at the cranial side, situated from the thoracic inlet to the superior mediastinum) with no connection to the cervical gland that seemed fine and was not enlarged. This was a 50-year-old female who was accidentally detected with an EMT after she had a 4-month history of cough. A CT scan showed a heterogeneous mass of 4 cm. The lady had normal thyroid function and negative antibodies against the thyroid. Before surgery, FNA for both thyroid sites showed no malignant traits [80]. Similarly, Uchida et al. [59] reported an EMT resection via a trans-cervical approach that was feasible due to a high position and small size (of 1.5 cm) in an MALT-positive patient [59]. Regal et al. [85] introduced the case of a young lady with an EMT of 10 cm diameter upon resection by performing midline partial sternotomy; the mass was located in the upper part of the anterior mediastinum with tracheal compression from the left side [85]. Coskun et al. [53] published a study in 2014 comprising 665 subjects who underwent thyroidectomy, and 6.3% of them (N = 42 adults) had a substernal goiter; among this subgroup, only 9.5% (N1 = 4 individuals) had a median sternotomy plus cervical incision (and one of these four patients had an EMT, thus representing 2.8% of the subgroup identified as “substernal goiter”); otherwise, only a cervical approach was used in 90.5% of the studied group) [53]. The analysis of Aziz et al. [48] showed that all individuals with EMTs (7/7) underwent surgery. The decision was individualized depending on location within the mediastinum. Two out of the three subjects with EMTs at the upper mediastinal had trans-cervical surgery, and one of these three individuals had a sternotomy (in association with concomitant valve replacement). Four patients with EMTs located at non-upper compartments underwent EMT resection through the chest with robotic assistance or posterolateral thoracotomy [48].

Some reports revealed patients who underwent total thyroidectomy in their medical history (one to 39 years prior to the moment of EMT confirmation) [35,118,120,122]. This previous neck surgical history allowed a cervical incision-based EMT removal in some cases [118,120,122] or required an additional sternotomy due to the specific EMT location (and vessels configuration) or suspected EMT malignancy [35,51]. Santangelo et al. [52] also introduced a case series of individuals with total thyroidectomy and sternotomy [52]. As mentioned, synchronous total thyroidectomy, lymph nodes resection and sternotomy for papillary thyroid cancer in EMTs was also performed by Toda et al. [55]. Metere et al. [88] used the combination of thyroidectomy via a cervical approach and longitudinal sternal splitting to remove the EMT [88].

In addition to sternotomy, thoracotomy was required, too; we mention a mostly singular case of an EMT located within the posterior mediastinum in a subject with situs inversus totalis to whom benign EMT removal was performed via a left posterolateral incision. She had a history of left hemi-thyroidectomy 30 years prior for a toxic thyroid nodule (with hyperthyroidism on first admission) with consecutive normal thyroid function including at the moment of EMT diagnosis [124].

VATS has gained success in mediastinal and non-mediastinal conditions as part of the modern thoracic surgery due to its benefits such as better recovery, shorter hospital stay, and reduced blood loss. VATS is limited by the tumor size of less than 10 cm; it seems safe even in cases of EMT malignancy as proved by the case published by Caroço et al. [42]. VATS, since its introduction in the early 2000s for non-thyroid conditions such as lung tumors, extended its indications, and an EMT is a successful candidate to VATS, as similarly seen for other endocrine (such as mediastinal parathyroid tumors) and non-endocrine conditions [194,195,196,197]. For instance, uniportal VATS was used by El Haj et al. [78] in a case of a 59-year-old lady who was synchronously identified with a mediastinal and a lung mass. She had a 4-month history of respiratory complaints. CT showed a tumor in the upper right mediastinum displacing the upper cave vein and compressing trachea on the contra-lateral site. She also had a left basal lung mass. Bronchoscopy showed pulmonary actinomycosis; thus, she underwent a 6-week course of injectable penicillin followed by a 3-month oral regime with the pulmonary mass remission. After a mediastinoscopy, an EMT (weighting 32 g) was removed via the uniportal VATS [78]. Carannante et al. [84] also reported the use of uniportal VATS in 2019 for an EMT of 98 g (histological report showed a cystic component and hemorrhage) [84].

The recent introduction of robotic-assisted thoracoscopic surgery (RATS) for ectopic parathyroid tumors [198] was extended to EMTs. Generally, the upper mediastinum is a difficult site for thoracoscopy. The location of vessels and nerves adds a challenge to this type of approach. In the case reported in 2023, RATS was used in a 40-year-old male who was accidentally detected with a mediastinal mass following COVID-19 infection that required a CT scan. A tumor of 6.1 by 7 by 6.1 cm was detected in the upper mediastinum. The tumor displaced trachea to the left. Despite being apparently incidental, retrospectively, the patient showed prior unexplained dysphagia and thoracic (and cervical expansion) pain. Once suspected, the second imaging procedure was conducted in terms of having a ^131^iodine scintigraphy that confirmed an EMT and a normal cervical thyroid gland. Endoscopic ultrasound-guided biopsy was performed followed by RATS with cervical tumor extraction via cervicotomy. The 2-day hospitalization was followed by an asymptomatic recovery in this non-malignant EMT case [76]. Of note, the first robotic resection of a large cervical goiter with a synchronous EMT underlying an adenoma was reported in 2004–2005 by Bodner et al. [127,128]. This was a 72-year-old lady who underwent a thoracoscopic resection with a da Vinci robot [127,128].

To summarize, the highly customized surgical approach in EMTs showed a wide area of interventions: VATS [78,84,87,93,125]; RATS [76]; resection via mediastinoscopy [81]; thoracoscopy [7,82]; thoracotomy [22,45,62,119]; right lateral thoracotomy [74,100,104] plus total thyroidectomy [69]; left posterolateral thoracotomy for EMTs [124] plus pulmonary cancer removal [101]; lateral incision for thoracotomy associated with a modified trans-manubrium approach [70]; sternotomy with total thyroidectomy [52,55,117]; sternotomy with cervical incision [53,117] and prior thyroidectomy [120]; sternotomy plus valve replacement [48]; VATS and total thyroidectomy for thyroid cancer [42]; cervical incision [38,59,106] plus right thyroid lobe removal [98]; mid-sternal body excision with sternum reconstruction and total thyroidectomy [60]; excision of clavicular metastasis and total thyroidectomy with modified neck dissection [61]; EMT and thymus removal plus total thyroidectomy [64,94]; midline sternotomy [13,35,85]; partial upper sternotomy [87]; median sternotomy for intra-pericardial EMTs [29]; sternotomy and thymectomy in a patient with a prior history of thyroidectomy for cervical thyroid cancer [123]; left hemi-thyroidectomy and isthmectomy plus median sternotomy [129]; partial sternotomy and total thyroidectomy associated with a selective parathyroidectomy via mediastinal and cervical exploration [115].

### 2.10. Outcome after Identification of EMTs

Regardless of the pathological traits, awareness of EMTs is essential, while removal was decided in most of the cases (rather than conservative approach) depending on the location and EMT anatomical features; the risk of malignancy (the rate of conversion from benign to a malignant EMT is not clearly understood, especially in long standing goiter-like EMT); the ectopic tissue enlargement with compressive symptoms/signs such as respiratory obstruction or compression on mediastinal organs; the risk of hemorrhage [84]; the patient’s co-morbidities and medical/surgical history as well as the general health status [7,13].

The main elements when it comes to the end results and consecutive management upon EMT identification follow:

EMT removal: based on the published data, this is the preferred approach and it is expected to have a good outcome and no local recurrence after surgery.

EMT surveillance: conservative approach, for instance, after EMT confirmation amid FNA results [89] is less likely preferred.

Firstly, thyroid dysfunction impairs the surgical outcome; thus, medication with anti-thyroid drugs for newly detected hyperthyroidism [34] is mandatory, and a further decision to be taken during follow-up (for instance, propylthiouracil until thyroid function normalization followed by total thyroidectomy and EMT resection [121] or methimazol followed by EMT removal [122]).

A second surgical step after EMT resection involves a total thyroidectomy followed by radioiodine ablative therapy, TSH-suppressive thyroxine treatment, and lifelong thyroglobulin monitoring in cases with a differentiated thyroid cancer in EMT [35,123].

External beam radiation therapy was applied for bone metastases [61].

Thyrosine kinase inhibitors were proposed for metastatic, aggressive thyroid cancer [61].

Follow-up of the thyroid function after EMT removal is required in order to check for (transitory) early post-operatory hypothyroidism [73].

The associated (iatrogenic) hypothyroidism via thyroxine replacement is corrected in benign thyroid disease, too.

Of note, levothyroxine-based TSH suppression therapy instead of surgery for benign EMTs in euthyroid patients [102] was not successful, and they are no longer encouraged according to the modern endocrine perspective (as similarly seen in thyroid nodules of the eutopic gland in patients with normal baseline thyroid function [199]).

The longest period of follow-up since EMT diagnosis and resection according to the reported data is up to one to two years [29,53,69,70,87,103]. Alternatively, Gao et al. [7] showed that EMTs remained stable for a median follow-up of 39.5 months in three cases that were not removed upon cytological confirmation and following the resection of one benign EMT [7]. Also, a stationary EMT for one year was registered in another case; the decision of a conservative approach was based on EBUS-guided biopsy results [118].

### 2.11. Proposed 10-Item Algorithm of EMT Approach

There are current gaps in EMT management, making it a matter of personalized approach: a low index of suspicion; no specific criteria for providing biopsy and/or FNA, neither for opting in favor of EMT resection in each case; lack of standardized surgical approach (also, concerning additional eutopic thyroid gland removal), and the imperious need of a multidisciplinary team decision. According to the prior mentioned data [7,11,29,52,59,70,76,83,85,89,92,95,98,120], we propose a working algorithm in EMTs which stands on ten major points of a multidisciplinary approach in helping a personalized decision (of note, this is our interpretation and opinion based on the current literature evidence rather than a consensus guideline): Clinical presentation

This is non-specific; it is a matter of a multidisciplinary panel, and one subject may be initially admitted at various medical and surgical departments. There are no pathognomonic elements. Dyspnea, cough, dysphagia, and chest pressure are the most common features. One patient may be completely asymptomatic.

2.CT scan

This is the most important and the standard imagery assessment, and it mostly represents the starting point in EMT suspicion. Approximately one out of ten patients is detected with EMTs as an incidental finding for unrelated conditions. Intravenous contrast is recommended, but this comes with the pitfall of contra-indication to use iodine contrast in patients with uncontrolled hyperthyroidism.

3.Re-do anamnesis

This is essential step since incidental detection does not mean completely asymptomatic. Retrospective data collection showed intermittent dyspnea in some circumstances, night sweats, facial edema, etc. These are expected to remit upon EMT removal.

4.Iodine or 99m-Tc scintigraphy

This investigation is pivotal for the thyroid tissue identification. Once an EMT is suspected, this becomes a mandatory step in diagnosis management. The potential pitfalls are the following: EMTs might have a different metabolic rate and a tracer uptake than a eutopic thyroid; some EMTs do not capture the tracer at all; the functional interplay might imply an over-activity of (autonomous) EMTs and consecutive cervical thyroid suppression or quite the other way; delayed captures in EMTs have been reported; prior use of iodine contrast or anti-thyroid drugs might block the iodine uptake; 99m-Tc might be up-taken in ectopic and eutopic parathyroid tissue as well.

5.Endocrine assessments

These are mandatory in terms of thyroid function (serum TSH, free thyroxine, and triiodothyronine), serum thyroid antibodies (antiTPO and antiTg), respectively TRAb—if hyperthyroidism is identified. For differential diagnosis, calcitonin (for medullary thyroid cancer), calcium and PTH values (for primary hyperparathyroidism), and paragangliomas-related assays are helpful. Hyperthyroidism requires the immediate starting of anti-thyroid drugs since this might complicate a biopsy but mostly a cervical thyroid or EMT manipulation during surgery. Thyroglobulin should be measured in any circumstance (at baseline and during follow-up) regardless of whether differentiated thyroid cancer is present.

6.Exploration of eutopic (cervical) gland and its connective tissue to EMTs

Thyroid ultrasound at the neck area is an easy-to-use and screening investigation. If thyroid nodules are identified, FNA should be completed according to the general guidelines. The connective tissue to an EMT makes it distinct from a retrosternal goiter, and it may be analyzed via imagery exploration (in addition to the intra-operatory findings and post-operatory histological report).

7.Second-line imagery

If prior investigations are not clear, further exploration becomes optional, not mandatory; for instance, SPECT-CT (for parathyroid tumours), MIBG scintigraphy (for paragangliomas), PET-CT (for a malignancy of any origin), and MRI of non-mediastinal areas might help (this depends of the particular circumstances in one case). Potential pitfalls in using PET-CT: some EMTs are hypermetabolic but not malignant; other EMTs do not uptake the tracer at all.

8.Biopsy or FNA in EMTs

This decision is individualized, and it is a matter of a multidisciplinary team based on the patient’s co-morbidities, EMT anatomical profile (including blood vessels configuration), clinical implications, access to safely perform a biopsy of FNA, and the patient’s preference. Potential pitfalls are represented by the fact that both types of investigations might be non-diagnostic (or provide results of uncertain significance). If the decision of EMT resection is already established, biopsy or FNA might not be necessary. Direct access to EMTs (a part from its resection) is mandatory in cases with prior or concurrent malignancies, whereas metastasis should be differentiated and the patient might not become a surgery candidate under these specific circumstances.

9.Surgical resection of EMTs

Surgery candidates mostly belong to one of these six categories: (1) suspected malignancy in EMTs (a rate of 18.8% was found); (2) compression symptoms (depending on anatomical effects and clinical features); (3) at-risk traits at imagery scans (e.g., necrosis, hemorrhage, increasing dimensions during follow-up); (4) lack of biopsy or FNA data (either they were not available, or they were non-diagnostic or the patient declined it, or they are too risky to be performed due to vessels’ proximity or due to the general health status of one patient and co-morbidities); (5) prior or synchronous non-thyroid cancers whereas the presence of a mediastinal metastasis might change the entire case management; (6) prior or synchronous thyroid cancer in the eutopic gland whereas surveillance based on serum thyroglobulin and whole body iodine scintigraphy cannot be adequately used due to the presence of the ectopic tissue. The type of surgical procedure is based on the decision of a highly skilled team of cardiothoracic surgery. Thyroidectomy, if malignancy is not suspected in EMTs or the cervical gland, is not necessary unless the EMT practical approach requires it. Minimally invasive procedures are the modern approach.

10.Long-term surveillance

Post-operatory assessment of the thyroid function is necessary since transitory hypothyroidism was reported. Iatrogenic permanent hypothyroidism requires long-term levothyroxine replacement. In cases with differentiated thyroid cancer, further total thyroidectomy (if was not already provided) is mandatory followed by the application of radioiodine ablative therapy, TSH-suppressive therapy with daily oral thyroxine, and the use of thyroglobulin lifelong periodical check-up. If the conservatory approach was chosen instead of EMT surgical resection, CT serial imagery scans are needed as well as the periodical thyroid function testing. In cases with prior thyroidectomy, a second histological analysis to compare the thyroid (cervical) tissue and EMT might help. Immunohistochemistry in EMTs may be necessary for differential diagnosis in selected cases (e.g., thyroglobulin, PAX8, TTF1, AE1/3, CD56, CD58, CD79a, Ck7, etc.). Further applications of molecular and genetic testing in EMTs are yet to be confirmed (Figure 4).

## 3. Discussion

### 3.1. Considerations on the Sample-Focused Analysis of EMTs

Our study on published data included 117 cases diagnosed with EMTs and identified a rather unexpected high rate of malignancy of 18.8% (underlying various histological profiles, but mostly papillary thyroid cancer) [7,9,11,13,22,29,34,35,36,37,41,42,45,48,49,50,51,52,53,54,58,59,60,61,62,63,64,65,66,67,68,69,70,71,72,73,74,75,76,77,78,79,80,81,82,83,84,85,86,87,88,89,90,91,92,93,94,95,96,97,98,99,100,101,102,103,104,105,106,107,115,116,117,118,119,120,121,122,123,124,125,126,127,128,129,130]. To our knowledge, this is the most complex analysis in EMTs that followed the clinical presentation, imagery, pathological traits, intervention, outcome, and potential pitfalls in addressing EMTs amid daily practice.

One of the most challenging aspects of EMTs includes the very first step of suspicion or recognition, since the clinical index is very low. We performed a secondary analysis upon primary data extraction regarding the findings at first admission for EMTs; they are not specific and belong to a very heterogeneous picture that might mimic multiple other conditions from cardiology, internal medicine, endocrinology, oncology or surgery domains [200,201,202,203,204,205,206]. The most frequent complaints involve sternal/chest pressure or pain, cough, dysphagia and dyspnea (Table 5).

The symptoms/signs were described as being acute (and required an admission as emergency [118]) or relatively recent (in terms of a few days [93], weeks [50], or months [9]) or they were presented in the medical history of one subject for previous 3 [74] to 5 years [86]. Hence, an EMT plays the role, as it has been described, of a great “imitator” [11] or “mimicker” [87] or the “forgotten goiter” [120].

When it comes to cancer in EMTs, the prior mentioned study of Santangelo et al. [52] identified an overall rate of 3.6% (less than our results) concerning malignancy in ectopic thyroid tissue, particularly, papillary thyroid carcinoma (N = 3092 patients who underwent a total thyroidectomy for any type of thyroid condition at eutopic gland). The study showed across five patients with EMT that one of them had this type of malignancy only in an ectopic rather than an orthotopic site (meaning one out the five EMTs was malignant, which is close to our results) [52]. For example, Caroço et al. [42] reported in 2023 a novel case of papillary thyroid carcinoma amid EMT recognition. This was a 73-year-old male who underwent total thyroidectomy for bilateral papillary thyroid carcinoma (no lymph nodes involvement) at the level of the orthotopic gland with post-operatory high serum thyroglobulin that required additional imaging exploration. Neck ultrasound and CT scan showed a mediastinal tumor of 6 by 3 cm (upper anterior compartment). At first, this was regarded as a potential metastasis from originating thyroid malignancy, which represents the stepping stone in cases with previous diagnosis of (orthotopic) thyroid malignancy. Resection via VATS was successful. A papillary micro-carcinoma was identified at EMT (0.4 cm). Post-operatory, the thyroglobulin remained undetectable for a post-operatory 6-month period (the patient did not received further radioiodine ablation) [42]. Another case published in 2012 showed two synchronous papillary thyroid malignancies within EMTs and a micro-carcinoma in the eutopic gland (multifocal carcinoma) [36].

On the other hand, Toda et al. [55] introduced a three-patient series with so-called “occult” thyroid carcinoma, and two out of the three subjects had a thyroid malignancy in the EMT as the primary origin not in the cervical gland. The first subject was a 71-year-old male with a 9-month history of superior vein cave syndrome (eyelid edema, jugular venous distension, and persistent facial edema). CT showed a lump of 2.7 cm in the upper mediastinum on the right side causing the stenosis of the upper cave vein. PET-CT confirmed this tumor in addition to multiple bone metastasis and normal thyroid findings. Bone biopsy confirmed a thyroid papillary malignancy with positive immunostaining for thyroglobulin and PAX8. Thyroid ultrasound and function were normal. Total thyroidectomy and radioiodine ablation were completed (100 mCI) yet with no specific radioiodine uptake within the mediastinum. The patient remained under surveillance without surgery for the mediastinal mass that was considered too risky due to the vessels’ proximity. There was also an 84-year-old male admitted for cough. A mass of 7.2 cm at the upper anterior mediastinum was identified at the CT scan (with tracheal compression and no connective tissue to the thyroid). EBUS-TBNA showed indeterminate results (no EMT confirmation). Thyroid function was normal with very high serum thyroglobulin (of 2450 ng/mL). The gentleman underwent combined total thyroidectomy, central neck dissection, as well as mediastinal tumor resection via sternal incision (sternotomy). The EMT showed a papillary thyroid cancer (no lymph nodes invasion); a 0.9 cm isthmus nodule showed a papillary micro-carcinoma that was considered by the authors as being a thyroid metastasis according to the post-operatory pathological analysis. Consecutive radioiodine ablation (100 mCi) was added [55].

On the contrary, Kesici et al. [35] reported in 2015 the case of a lady who underwent a total thyroidectomy for papillary thyroid carcinoma and post-operatory iodine scintigraphy highlighted EMTs, which was completely removed amid a second operatory procedure, but no malignancy in ectopic tissue was confirmed [35]. This was similar to the case introduced by Kamaleshwaran et al. [123], also revealing the importance of iodine scintigraphy to identify non-connected mediastinal tissue that does not host similar malignancy traits with the eutopic gland [123].

Additionally, Shafiee et al. [64] also reported a papillary thyroid carcinoma in an EMT that was initially suspected as being a thymus malignancy due to its location; additionally, the eutopic thyroid showed no carcinoma [64]. Karim et al. [60] showed the finding of an EMT in a 55-year-old female underlying follicular variant of papillary thyroid carcinoma. The presentation was with a progressive tumor development amid the latest two-year period. She had a history of a right lobectomy for a benign thyroid condition with consecutive normal thyroid function, including on admission. The EMT was located within the anterior compartment (of 15 by 5 cm at CT scan) without any associated malignancy at the level of ectopic gland remnants. The lady underwent EMT removal (mid-sternal body excision with sternum reconstruction) after having a trunk biopsy that confirmed an EMT via positive TTF1 stain associated with total thyroidectomy followed by radioiodine ablative therapy (80 mCi). The initial serum thyroglobulin was very high (>6000 ng/mL). In addition, whole body scan showed several metastases [60]. Hu et al. [62] showed in 2017 primary papillary thyroid cancer features in EMTs based on a cytological report that was provided by EBUS-TBNA in a male senior later confirmed after tumor removal as having no malignancy in the eutopic thyroid gland [62]. A dramatic case was reported in an adult lady who first showed a right distal clavicular mass of 5.5 cm that turned out to be a metastasis from an EMT with primary papillary thyroid cancer and no malignancy within the eutopic gland, which was small and atrophic. FNA at clavicle mass confirmed a thyroid malignancy (positive for TTF1, thyroglobulin). ^18^F-FDG PET-CT showed multiple hyper-metabolic lymph nodes and EMT uptake (that displaced trachea and esophagus). The 68-year-old female underwent excision of the right clavicular mass followed by total thyroidectomy plus modified right neck dissection (levels 3, 4, and 5) followed by external beam radiation therapy to the distal clavicular region. Central neck biopsy of the superior mediastinal mass confirmed papillary thyroid cancer. She underwent radioiodine therapy; then, CT showed multiple metastatic lymph nodes the neck and upper mediastinum and additional pulmonary spreading. After additional radioiodine therapy and seemly remission, further recurrence was noted at PET-CT scan, and a neoplasia switch to a dedifferentiated type required therapy with a thyrosine kinase inhibitor. This unusual aggressive form of thyroid malignancy was considered at the moment of publication (in 2018) the most severe outcome in EMTs at that point [61].

### 3.2. EMT Submitting to the Scenario of an Endocrine Incidentaloma

Approximately one out of ten patients diagnosed with an EMT was accidentally detected via various imagery assessments, mostly CT scan, which were completed for different purposes (related to malignant or benign conditions) [7,11,29,52,59,70,76,83,85,89,92,95,98,120]. Awareness represents the key operating factor, since the EMT is situated at the crossroads between mediastinal masses with oncologic background and endocrine incidentalomas (in everyday endocrine practice, the traditional incidentalomas involve thyroid, adrenal and pituitary glands) [90,207,208]. Notably, one 42-year-old male was admitted for EMT-related complaints, and finally, he was identified with a right adrenal tumor of 5 cm (an adrenal incidentaloma) [73]. So far, identifying an adrenal tumor in an EMT patient seems like an accidental finding, and further research is needed to pinpoint a potential pathogenic connection. Nevertheless, the adrenal gland was exceptionally found to be involved as a host of an ectopic thyroid tissue despite the fact that the embryological rationale behind this is less understood [209,210].

### 3.3. The Impact of COVID-19 Pandemic in Reporting a Mediastinal Thyroid Tissue

As mentioned, some patients diagnosed with EMTs presented fever or sub-fever, and this required a differential diagnosis with an infectious disease [13,102]. Generally, many imagery investigations, particularly in the area of lung, mediastinum and cervical, were performed during or after one individual suffered a coronavirus infection during COVID-19 pandemic years, and many novel entities were described, or prior unknown conditions of one subject were detected under these circumstances while not being related to the viral infection itself [211,212,213,214,215]. For instance, a 40-year-old male was found with a 6 cm EMT amid CT scan for COVID-19 infection (the case was published in 2023) [76]. On the other hand, some authors suggested that restrictions during pandemic waves limited the access to prompt intervention in some instances, whereas an ectopic thyroid was involved [8]. Whether practice amid COVID-19 pandemic years stands as a pillar for a more frequent EMT detection or for a delay of prompt EMT recognition is yet to be proven. We performed an additional analysis on the papers with regard to the EMT diagnosis [7,11,29,52,59,70,76,83,85,89,92,95,98,120] and found that 26% of articles (regarding EMTs) were published between 2020 and the present time, representing almost half of the original articles published between 2000 and 2019. We expected an expansion of this specific topic under the umbrella of increasing awareness and elevated access to various imagery assessments (Figure 5).

### 3.4. Ectopic Parathyroid versus Thyroid within Mediastinum

As mentioned, EMTs should be differentiated from ectopic parathyroid tumors. 99m-Tc sestamibi scintigraphy might identify parathyroid tumors (including at mediastinal levels). This is mostly a late capture, while the early tracer uptake may be found in eutopic and heterotopic thyroid tissue, and this aspect might be a potential source of bias in this particular instance [216]. In case of functional parathyroid tumors, additional blood assays offer the biological confirmation of primary hyperparathyroidism (high serum calcium and PTH), and further exploration of associated complications such as bone disease (including osteoporosis) or kidney anomalies is mandatory [217,218,219,220]. Of note, the interplay between ectopic thyroid and ectopic parathyroid tissue might be related to the embryological origin; as mentioned, the thyroid originates from the first and second pouches, while superior parathyroid glands come from the third pouch, and inferior parathyroid glands originate from the fourth pouch [216,219]. CT scan in addition to 99m-Tc sestamibi scintigraphy has a very high sensitivity (100%) and positive predictive value (97%) for parathyroid tumors, including for the ectopic masses. A lower rate of detection is found in small tumors with a less mitochondrial count that relates to a decreased tracer uptake [216]. Recent data agree that even better results are expected for using SPECT-CT and combined techniques such as dual-phase 99m-Tc MIBI scintigraphy and 99m-Tc MIBI SPECT-CT [216,217,218,219,220]. Approximately 3% to 16% of the population has a supra-numerary parathyroid gland (from five to eight accessory parathyroid glands), and this may be an issue of ectopic parathyroid tumor identification in cases with associated primary hyperparathyroidism that need tumor removal or in subjects with recurrent/persistent hyperparathyroidism upon the parathyroidectomy of orthotopic parathyroid tumors [216,217].

Notably, 60% of the supra-numerary parathyroid glands are in the mediastinum, and the most frequent site is the thymus [216,217,218,219,220]. For example, Muzurović et al. [115] introduced an interesting case of a 53-year-old women admitted for a 3-month history of cough and chest pain. CT scan showed an upper anterior mediastinal mass of 9.5 by 7.5 by 11.5 cm (with heterogeneous structure, central calcifications with a mild displacement of the thoracic aorta, pulmonary artery and trachea). CT-guided biopsy confirmed an EMT. Additionally, neck ultrasound showed a dominant thyroid nodule at the left lobe. FNA showed no malignancy. The initial hormonal panel was normal for the thyroid profile, but high serum calcium (of 3.11 mmol/L; normal ranges between 2.2 and 2.55 mmol/L) and PTH (of 152 pg/mL; normal ranges between 10 and 65 pg/mL) represented the biological confirmation of a primary hyperparathyroidism. The second imagery procedure, namely, 99m-Tc pertechnetate scintigraphy, identified the tracer uptake at the eutopic thyroid and the EMT. 99m-Tc sestamibi scintigraphy with SPECT-CT scan showed a right inferior parathyroid tumor and another ectopic parathyroid tumor (on the left side of the upper third of mediastinum). The complex surgical procedure included resection of an EMT with total thyroidectomy, selective parathyroidectomy and thymectomy via mediastinal and cervical exploration (partial sternotomy and Kocher cervicotomy). Post-operatory confirmation of benign features included a normal thyroid gland, colloid goiter with cystic transformation in a giant EMT plus the two parathyroid adenomas. Post-operatory PTH remained normal after the first day. Iatrogenic hypothyroidism required levothyroxine replacement. A good outcome was registered after 3 months of follow-up. Most probably, a similar embryological source might explain the co-presence of these two types of ectopic tissues, albeit this would be exceptional [115].

### 3.5. Limitations and Further Expansion

Our research is a non-systematic review across a single database; however, this is the largest sample-based analysis of published data when compared to prior publications. The level of statistical evidence and the heterogeneous spectrum of provided data varied, and we did not intend to limit the inclusion of interesting cases with various presentations over the years.

The topic of disorders involving thyroid morphogenesis poses some issues that are still a matter of debate in the field of EMTs such as the rate of malignant transformation in long-standing EMTs; the risk of further EMT enlargement and hemorrhage as well as the expected annual rate of growth; the importance of conducting a pre-operatory histological and/or cytological report; the best imagery option and the optimum interventional strategy as well as the selection criteria for surgery candidates. Moreover, we used the original terms regarding the pathological report, which is very important in malignant EMTs, but changes of nomenclature and classifications were registered amid this search of more than three decades [216,217,218,219,220]. Another topic that we anticipate will gain increasing interest is represented by the molecular and genetic exploration in subjects confirmed with EMTs. In our analysis, *SQSTM1-NTRK3* chromosomal rearrangement was identified in one case [58], and another patient with anaplastic carcinoma harbored a *BRAF* pathogenic variant [37]. As mentioned, a single pediatric case was found [68], but generally, the connection between the endocrine disruptors in terms of environmental factors (such as the geographic influence, the air pollutions, etc.) or internal factors like infectious conditions (as recently shown by the COVID-19 pandemic) or autoimmune (namely, endocrine and non-endocrine autoimmunity) issues should be studied in relationship with the developmental conditions, specifically, the ectopic thyroid (either acting during pregnancy or early during childhood); yet this represents a future topic to be studied [221,222,223,224,225,226]. Further on, the exact impact on the overall thyroid cancer-associated burden with respect to ectopic thyroid-related malignancy represents an open issue according to the current level of statistical evidence; we expect a future expansion amid increased awareness and access to advanced imagery tools and performant surgical access with an overall good long-term outcome in ectopic thyroid cases [227].

## 4. Conclusions

EMT stands for one of the most fascinating and challenging medical and surgical entities; a multidisciplinary team is mandatory. The sample-based analysis identified 117 patients with EMTs, and 18.8% of them displayed a malignancy in the EMT. The most common histological type was papillary thyroid cancer. While most cases had a good outcome, emergency admission due to local compressive effects needs a life-saving intervention; on the other hand, a very aggressive fulminant evolution has been reported in a small number of patients with follicular-cell-derived carcinoma. One of the most striking aspects in analyzing EMTs involves a wide area of imaging procedures but mostly of biopsies/FNA and surgical interventions. Despite the lack of a standardized approach, the customized management may be synthetized in a 10-item algorithm as we mentioned. CT scan remains the major imagery element, and iodine or 99m-Tc scintigraphy represents the most practical one once the ectopic intra-thoracic tissue is suspected. An endocrine panel is required at any point from baseline to long-term follow-up. The selection of biopsy or FNA methods varies depending on EMT profile, co-morbidities and available procedures. A skillful surgical team for EMT removal with or without thyroidectomy offers a very good prognostic in majority of EMT patients.

## Figures and Tables

**Figure 1 cancers-16-01868-f001:**
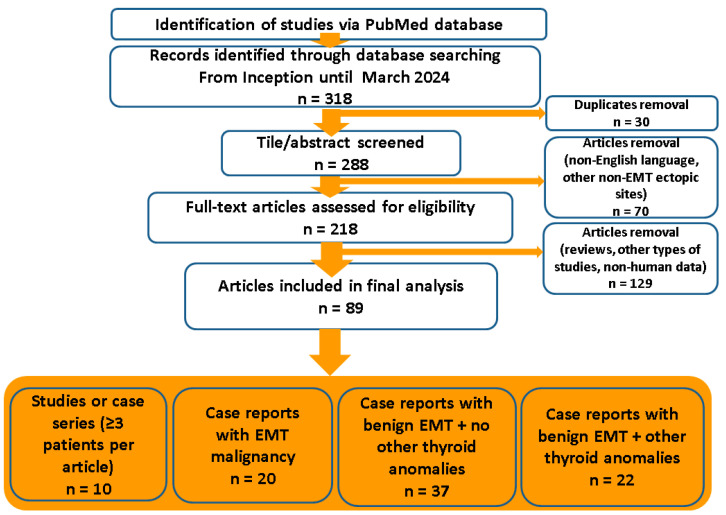
Diagram flow of search according to our methods. Abbreviations: EMT = ectopic mediastinal thyroid; n = number of papers.

**Figure 2 cancers-16-01868-f002:**
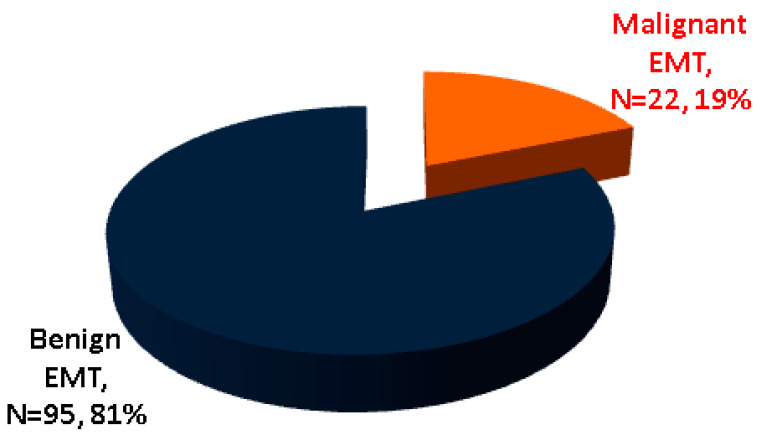
The malignant profile in EMT according to our analysis (N = number of patients) [36,37,41,42,55,58,59,60,61,62,63,64,65,66,67,69,70,71,72].

**Figure 3 cancers-16-01868-f003:**
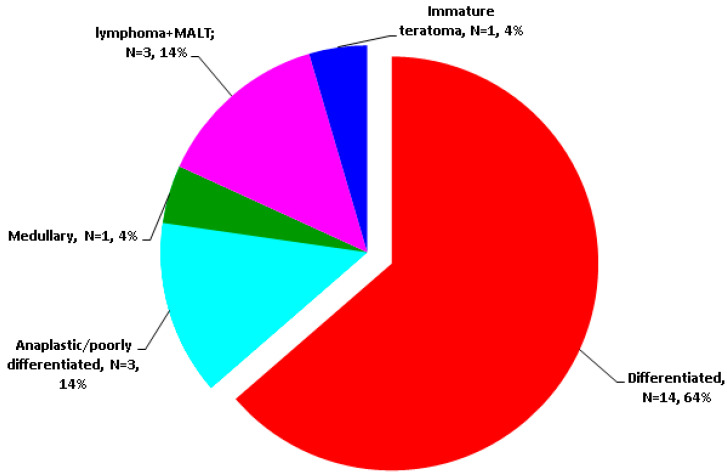
Cancer-focused analysis amid our EMT research [36,37,41,42,55,58,59,60,61,62,63,64,65,66,67,69,70,71,72] (N = number of patients).

**Figure 4 cancers-16-01868-f004:**
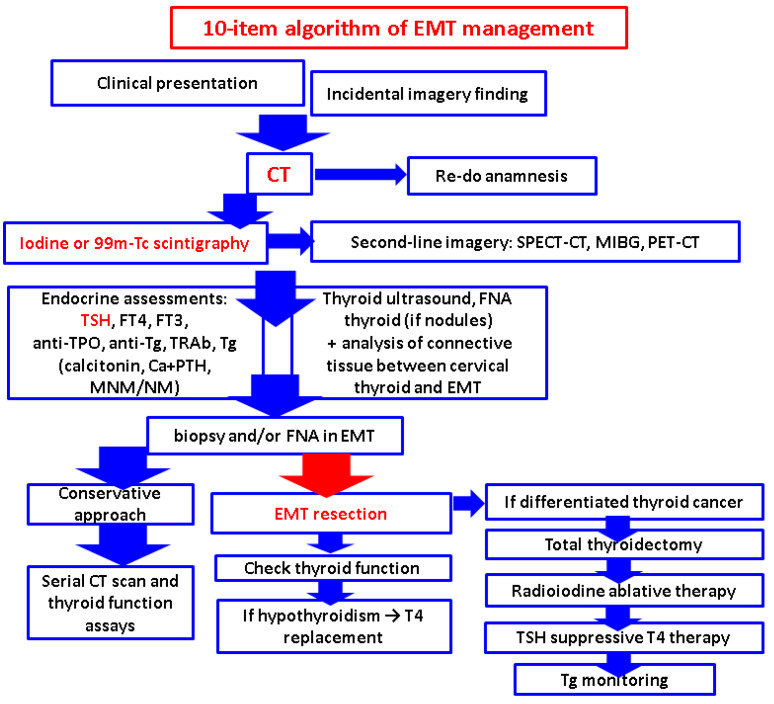
Proposed 10-item algorithm of EMT management: a multidisciplinary perspective of an otherwise customized decision. Abbreviations: antiTPO = anti-thyroperoxidase antibodies; antiTg = anti-thyroglobulin antibodies; CT = computed tomography; Ca = calcium; EMT = ectopic mediastinal thyroid; FNA = fine needle aspiration; FT4 = free thyroxine; FT3 = free triiodothyronine; MIBG = metaiodobenzylguanidine; MNM = metanephrines; NM = normetanephrines; PTH = parathormone; PET-CT = positronic emission tomography-computed tomography; SPECT-CT = single-photon emission computed tomography; Tc = technetium; TSH = thyroid-stimulating hormone; TRAb = TSH receptor antibody); Tg = thyroglobulin.

**Figure 5 cancers-16-01868-f005:**
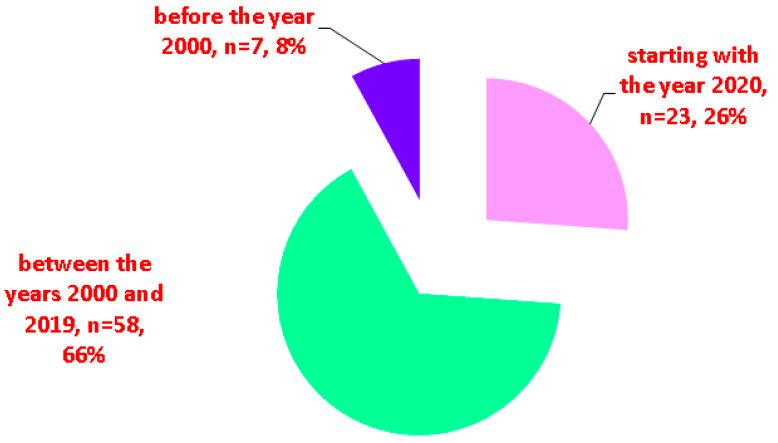
The perspective of publication timeline in EMT according to our methods (n = number of original publications).

**Table 1 cancers-16-01868-t001:** Studies or case series that analyzed patients with EMT among other outcomes; the display starts with the most recent publication date [7,9,45,48,49,50,51,52,53,54].

First AuthorYear of PublicationReference	Studied Population Pathological ReportConnective Tissue with Cervical Thyroid (If Specified by the Original Authors)	Clinical DetectionEndocrine Panel (Thyroid Function)Data on Orthotopic Thyroid (If Specified by the Original Authors)	Imaging TraitsOther Assessments That Were Applied for Diagnosis (If Specified by the Original Authors)	Biopsy (If Specified by the Original Authors)SurgeryOutcome
Aziz2023(*)[48]	 retrospective analysis(between 1996 and 2021)  Standford database (high-volume academic surgical center)  7 EMTs/202 ectopic thyroids  100% (7/7) benign thyroid follicular epithelium  0% (0/7) no connection with ectopic thyroid	 6/7 patients: compressive symptoms or hyperthyroidism  1/7 patient: incidental finding at CT  4/7 patients had endocrine assessments (4/4 patients had normal thyroid function)  7/7 patients with normal cervical thyroid	 7/7 patients had a CT scan  3/7 patients: superior compartment (A1)  1/7 patient: middle compartment (A2)  2/7 patients: posterior compartment (A3)  1/7 patient: anterior compartment (A4)  3/7 patients had a second imaging procedure:  1/7 patient: PET  1/7 patient: radioiodine scan  1/7 patient: cervical MRI	 no data on pre-operatory biopsy  7/7 patients had surgery: A1 (N = 3):  2/3 patients with cervical approach  1/3 patient with sternotomy (+valve replacement)non-A1 (N = 4)  resected through the chest with robotic assistance (2/7) or posterolateral thoracotomy (2/7)
Motlaghzadeh(*)2023[9]	 case series of 7 EMTs  F/M = 4/3 (mean age of 54 y)	 2/7 patients had cough and chest pressure  1/7 patient had a 6-month dyspnea and dysphagia  1/7 patient had intermittent night sweats	 mean EMT diameter of 3.6 cm (ranges: 2–6 cm)	 before surgery:  1/7 patient had mediastinoscopy  1/7 patient had EBUS-guided biopsy  surgery (7/7)  mean hospital stay of 4.3 days  1/7 patient had a post-operatory complication: chyle leak & phrenic nerve injury (conservative approach)
Gao2023[7]	 retrospective, single center study in ectopic thyroid (N = 47) (between 2013 and 2022)  rate of EMT:4/47  F/M: 3/1 (mean age of 55.5 y)  100% (4/4): benign EMT  1/4 patient had a synchronous thymus lipoma	 3/4 patients: incidental detection  1/4 patient: had a detection due to cough  4/4 patients had normal thyroid function  4/4 patients with normal cervical thyroid	 4/4 patients had a CT scan:  3/4 patients: upper compartment (A1)  1/4 patient: posterior compartment (A2)	 A1: FNAC → follow-up (no surgery)  A2: thoracoscopic resection  A2: stable during follow-up → no post-operatory hypothyroidism (A2)  median follow-up: 39.5 months
Vuorisalo2023[49]	 case series of quality assurance programme in EBUS-TBNA (N = 3)  F/M = 2/1  ages: 86 (F1, F2), 81 (M) y  mediastinal lymph nodes enlargement (EMT) in 3 patients	 F1: thyroid enlargement → CT scan → identification of mediastinal lymph nodes → EBUS-FNA (via esophagus) suggested EMT  F2: thyroid enlargement → CT scan → identification of mediastinal lymph nodes → EBUS-FNA (via esophagus) suggested EMT  M: previous history of lung adenocarcinoma → linear-EBUS to rule out malignancy spreading → EMT	-	-
Ozturk2018[50]	 case series of 3 EMTs using EBUS-FNA  F/64 y (F1)  F/53 y (F2)  M/40 y (M)  benign EMT	 F1: history of total thyroidectomy of multinodular goiter  F1: asymptomatic  F2: 1-month cough  M: 2-month dyspnea	 F1: CT: EMT of 8 × 5.2 cm  F2: CT: EMT of 5.5 × 4.2 cm  M: CT: EMT of 6.3 × 3.6 cm	 EBUS-TBNA confirmed EMT upon cytological report
Sunam2016[45]	 case series of 8 patients with surgery for huge mediastinal masses of different types  M/F = 2.6 (mean age of 34.5 y)  1/8 patient had EMT (M/44 y → benign EMT	 presentation with hemoptysis, orthopnea	 CT exam (emergency): EMT of 12 × 7 cm  EMT site: upper-middle mediastinum	 thoracotomy
Di Crescenzo2016[51]	 single center, retrospective study on surgery for cervico-mediastinal goiters (between 1991 and 2006)  N = 97 patients who underwent surgery for cervico-mediastinal goiters		 49.2% patients had a cervico-mediastinal goiter  40% patients had a mediastino-cervical goiter  10.8% patients had a mediastinal goiter	 22 patients had (total or split) sternotomy  11 of them had “forgotten” goiter in mediastinum (no specific data of EMT)
Santangelo2016[52]	 retrospective study in ectopic thyroid on patients who underwent thyroidectomy (between January 2000 and December 2013)  N = 3092 patients  N1 = 28 patients with ectopic thyroid (0.9%) Location:  21.4%: lateral cervical  21.4%: thyroglossal duct  17.9%: EMT (N = 5 patients)  17.9%: lingual  10.7%: sublingual  10.7%: submandibular  EMT: F/M = 3/2 (age range:36–51 y; mean age of 41 y)  overall rate of malignancy in all ectopic thyroid tissue: 3.6% (papillary thyroid carcinoma)  4/5 patients with EMT had multinodular goiter in EMT  1/5 patient with EMT had nodular goiter and a papillary thyroid carcinoma in EMT	 4/5 patients with EMT had multinodular goiter (eutopic)  1/5 patient had congenital hypothyroidism treated with levothyroxine(presentation of EMT for 1-y history of dyspnea)—C1  2/5 patients had incidental EMT detection (asymptomatic)  2/5 patients had mediastinal compression syndrome  thyroid function:  hypothyroidism (C1)  2/5 patients with normal thyroid function  2/5 patients with hyperthyroidism	 C1: EMT compression on trachea + EMT confirmation via scintigraphy  iodine scintigraphy was performed in all 5/5 cases	 4/5 patients had sternotomy + total thyroidectomy
Coskun2014[53]	 retrospective study (between 2005 and 2012)  N = 665 patients who underwent thyroidectomy  N1 = 42 (6.3%) patients with substernal goiter  F = 62% (mean age = 50 y, range: 24–48 y)  M = 38% (mean age = 53 y, range: 18–76 y)  1/4 patient had EMT		 CT, MRI and scintigraphy as pre-operatory assessments	 N2 = 4(9.5%) patients required a sternotomy + cervical incision(including the patient with EMT)  N1: average post-operatory follow-up: 2 y (range: 6–40 months)
Kozol1993[54]	 N = 16 patients with benign ectopic thyroid tissue (lateral aberrant type)  N = 1/16 patient had EMT	 N = 15/16 patients had benign thyroid nodules	 N = 7/16 patients were detected amid investigations for primary hyperparathyroidism  N = 9/16 patients were detected amid investigations for cervical thyroid diseases	 N = 8/16 patients had thyroid resections for occult malignancy (that were not confirmed after surgery)

Abbreviations: EBUS = endobronchial ultrasound; EBUS-FNA = endobronchial ultrasound-guided fine needle aspiration; EMT = ectopic mediastinal thyroid; EBUS-TBNA = endobronchial ultrasound-guided transbronchial needle aspiration; F = female; FNAC = fine needle aspiration cytology; M = male; MRI = magnetic resonance imaging; PET = positronic emission tomography; y = years; (*) these were two studies exploring the same database, hence the same population with EMT; MRI (blue font = patients with EMT).

**Table 2 cancers-16-01868-t002:** Case reports of EMT underlying different types of malignancy according to our methods; the display starts with the most recent publication date [36,37,41,42,55,58,59,60,61,62,63,64,65,66,67,68,69,70,71,72].

First AuthorReference Number	Publication Year	Sex	Age(y)	Pathological ReportConnective Tissue with Cervical Eutopic Thyroid (If Specified by the Original Authors)	Clinical DetectionEndocrine Panel (Thyroid Function)Data on Orthotopic Thyroid (If Specified by the Original Authors)	Imaging Traits of EMTOther Assessments That Were Applied for Diagnosis (If Specified by the Original Authors)	EMT Biopsy or FNA (If Specified by the Original Authors)Surgery for EMT	Outcome
Khthir[58]	2024	F	67	 EMT: metastatic papillary thyroid cancer (focal necrosis, extensive invasion into the fibro-adipose tissue, invasion of the vagus nerve)  benign orthotopic thyroid  no connection to orthotopic gland	 admission for distant metastasis (thoracic T2 and T3 spine causing cord compression)—upper back pain for the last few weeks  normal thyroid function  a thyroid nodule of 1.2 cm at ultrasound (post-operatory benign features)  thyroglobulin: 15,272 ng/mL (baseline) → 1857 ng/mL (post spine, thyroid and EMT surgery, spine radiotherapy, radioiodine therapy)	 MRI: vertebral metastasis confirmation  CT: EMT of 3 × 2.5 × 2.5 cm  EMT site: left upper mediastinum with extension to supraclavicular area  FDG-PET: hypermetabolic lesion in EMT  molecular testing in FNA: *SQSTM1NTRK3* chromosomal rearrangement	 laminectomy + decompression surgery at T1-T3 → diagnosis of papillary thyroid cancer → external beam radiation (spine)  FNA in EMT: papillary cancer  total thyroidectomy → benign thyroid  removal of EMT	 post-operatory 100 mCI → whole body scan: no metastasis → no “ gross ” recurrence after 14 months since admission
Nguyen[37]	2023	M	90	 anaplastic carcinoma due to a BRAF V600E pathogenic variant	 rash at sternal level  severe weight loss  prior treatment for unrelated hypothyroidism (for non-surgical causes, no specific cause is mentioned) and prostate carcinoma  atrophic thyroid (eutopic) gland at CT scan	 CT: EMT of 12 × 11 cm (rapid growth) + sternal skin and bone metastases  PET-CT after biopsy report showed lymph nodes metastases (cervical and axillary)	 skin biopsy → anaplastic + papillary component diagnosis → excisional biopsy	 6-month survival
Caroço[42]	2023	M	73	 multiple hyperplasic nodules  papillary micro-carcinoma (0.4 cm)—no peri-neural or vascular invasion (pT1aNxMx)  no anatomic connection to eutopic gland	 total thyroidectomy for bilateral papillary thyroid cancer → post-operatory persistent thyroglobulin → CT exam  post-thyroidectomy status on first suspicion of EMT  eutopic thyroid with thyroid cancer synchronous with thyroid cancer in EMT	 CT: EMT of 6 × 3 cm	 resection via VATS  EMT vascularization was from intrathoracic vessels	 2-day hospitalization amid VATS  no radioiodine-ablation  post-operatory 6-month follow-up: no recurrence
Uchida[59]	2020	F	67	 MALT in EMT (1.5 × 1.2 × 0.9 cm)  chronic thyroiditis in EMT + orthotopic thyroid  no connection to eutopic gland	 incidental detection  normal thyroid function  chronic thyroiditis in cervical thyroid (not MALT)  serological confirmation of autoimmune thyroiditis:  antiTPO = 33 UI/mL(0–27)  antiTg = 210 UI/mL(0–15.9)	 CT: EMT of 2 × 1.3 cm  site: upper mediastinum (anterior to trachea)  MRI  ^123^I-mIBG: excluded paraganglioma	 EMT resection via trans-cervical incision Post-operatory:  FNA of cervical thyroid  FDG-PET	no other therapy
Toda[55]	2020case 1	M	71	 metastatic papillary thyroid carcinoma in EMT  bone metastases with positive thyroglobulin and PAX8	 9-month history of superior cave vein syndrome  normal thyroid function  normal thyroid ultrasound	 CT: EMT of 2.7 cm  site: upper mediastinum (on the right side)  PET-CT: bone metastases and normal eutopic thyroid	 total thyroidectomy	 radioiodine ablation (100 mCi)→ continue surveillance
2020case 2	M	84	 papillary thyroid carcinoma in EMT  papillary micro-carcinoma in eutopic thyroid  no connection to eutopic gland	 cough  normal thyroid function  very high serum thyroglobulin (2450 ng/mL)	 CT: EMT of 7.2 cm  site: upper anterior mediastinum (tracheal compression)	 EBUS-TBNA in EMT: non-diagnosis  total thyroidectomy + central neck dissection + sternotomy	 radioiodine ablation (100 mCi)→ continue surveillance
Karim[60]	2018	F	55	 follicular variant of papillary thyroid carcinoma  no connection to eutopic thyroid	 2-year history of progressive mediastinal nodule  prior history of right thyroid lobectomy (benign)  normal thyroid function  very high pre-operatory thyroglobulin (of 6000 ng/mL)  normal eutopic tissue	 CT: EMT of 15 × 5 cm  site: anterior mediastinum  bone scintigraphy: bone metastases	 trunk biopsy: EMT confirmation  mid-sternal body excision with sternum reconstruction + total thyroidecotmy  vessels from bilateral internal mammary arteries  intra-operatory:bleeding from both internal thoracic arteries	 radioiodine therapy (80 mCi) after 6 weeks since surgery
Vázquez[61]	2018	F	68	 metastatic, aggressive papillary thyroid carcinoma  no connection to eutopic thyroid	 clavicular metastasis at onset (5.5 cm)  normal thyroid function  normal eutopic tissue	 FNA clavicle: confirmation of metastasis from papillary thyroid cancer  ^18^F-FDG PET-CT: lymph nodes metastasis and EMT	 excision of clavicular mass + total thyroidectomy + modified neck dissection	 external beam radiation therapy for clavicle→ radioiodine therapy → dedifferentiation→ thyrosine kinase inhibitor
Hu [62]	2017	M	76	 papillary thyroid carcinoma in EMT	 history of 1-month hoarseness  normal eutopic thyroid (pathological report)	 CT: 2.5 × 3.4 cm  site: tumor was adjacent to the central airway; contact with esophagus	 EBUS-TBNA: confirmed the thyroid cancer in EMT  tumor resection via thoracotomy  intra-operatory: EMT surrounded the recurrent laryngeal nerve + invaded the esophageal muscle  +total thyroidectomy	
Wu [63]	2015	M	54	 primary mediastinal large B-cell lymphoma	 1-month history of dyspnea and progressive neck mass increase  prior renal recipient under cyclosporine + mycophenolate mofetil for 11 y  Hashimoto’s thyroiditis diagnosed 3 y prior	 CT: neck mass + upper mediastinal mass	 surgery for neck mass → lymphoma confirmation	 the role on immunosuppressive medication
Shafiee[64]	2013	F	39	 papillary thyroid carcinoma in EMT (intra-thymus)	 2-month history of left hemi thorax pain  normal eutopic gland	 HRCT: intra-thymus mass of 8 cm  bronchoscopy: external compression due to EMT	 total thyroidectomy + thymus removal → confirmation of malignant EMT, no thymus malignancy	
Piciu][36]	2012	F	36	 EMT with papillary multifocal thyroid carcinoma	 eutopic thyroid: autoimmune thyroiditis + papillary micro-carcinoma (of 2 mm) in eutopic gland	 ^131^iodine whole body scintigraphy sodium iodide		
Wang [65]	2011	M	49	 medullary thyroid carcinoma in EMT	 admission: severe hypokalemia  ectopic ACTH syndrome	 CT: EMT of 5 × 5 × 5 cm  site: anterior mediastinum	 total thyroidectomy with central compartment and ipsilateral modified radical neck dissection	 post-operatory remission of ectopic ACTH syndrome
Camargos[66]	2010	F	95	 necropsy:anaplastic thyroid carcinoma + normal eutopic thyroid (absence of infectious disease)	 8-y history of dysphagia  on admission: eosinophilia, leukocytosis (leukemoid reaction + peripheral hyper-eosinophilia)  the patient died within 11 day due to respiratory failure	 EMT in upper mediastinum		
Demirag[67]	2009	M	80	 primary ectopic thyroid B cell lymphoma in EMT  no connection to eutopic gland		 site: anterior mediastinum		
Ranaldi[68]	2009	M	18 days	 mediastinal immature teratoma in EMT	 new born patient (singular case)	 upper mediastinal EMT	 EMT resection	 disease free for 22-months (post-operatory)
Shah[69]	2007	F	45	 EMT: colloid goiter with foci of follicular variant of papillary carcinoma  no connection to eutopic gland  eutopic thyroid: no malignancy	 2-month history of cough + dyspnea  normal thyroid function  increased thyroglobulin at baseline: 508 ng/mL	 CT site: lateral to the right atrium  99m-Tc scintigraphy: EMT uptake and eutopic thyroid	 FNA cytology in thyroid: follicular cells + a single group of atypical cells  CT-guided core biopsy showed scanty thyroid follicles with colloid  total thyroidectomy + EMT excision via right lateral thoracotomy	 radioiodine ablative therapy after surgery + T4 therapy  stable disease after 14 months since surgery
Yoshino[70]	2006	M	49	 EMT: large cyst + nodules underlying a thyroid carcinoma  no connection to eutopic gland	 incidental detection	 CT: EMT of 6.5 × 5.5 cm  EMT site: upper, anterior mediastinum (adherent to vertebrae, esophagus, trachea, and upper cave vein)	 lateral incision for the thoracotomy + modified trans-manubrium approach → EMT excision	 14-month post-operatory asymptomatic outcome
Zagar[71]	2003	M	34	 columnar cell carcinoma (rare variant of papillary carcinoma) in EMT	 rapidly growing neck mass, extending to the anterior and middle mediastinum	 site: anterior mediastinum (tracheal + oesophageal deviation)  99m-Tc scintigraphy  ^123^iodine scintigraphy  octreoscan	 near total thyroidectomy + lymph-node dissection	 post-operatory chemotherapy and radioiodine ablation
Dominguez-Malagon[72]	1995	F	64	 EMT: poorly diferentiated (insular) thyroid carcinoma		 EMT site: anterior mediastinum		
Mishriki[41]	1983	NA	NA	 EMT: Hürthle-cell thyroid tumor (malignant)	 clinically malignant (lung metastases)		 diagnosed based on FNA	

Abbreviations: ACTH = Adrenocorticotropic Hormone; antiTPO = anti-thyroperoxidase antibodies; antiTg = antithyroglobulin antibodies; cm = centimeter; CT = computed tomography; EBUS-TBFNA endobronchial ultrasound-guided transbronchial needle aspiration; EMT = ectopic mediastinal thyroid; FDG PET-CT = fludeoxyglucose positronic emission tomography-CT; FNA = fine needle aspiration; F = female; HRCT = High Resolution CT; MRI = magnetic resonance imagery; MALT = mucosa-associated lymphoid tissue; ^123^I-MIBG = iodine meta-iodobenxylguanidine; M = male; NA = not available; Tc = technetium; T4 = thyroxine; VATS = video-assisted thoracic surgery; red font = the type of malignancy in EMT.

**Table 3 cancers-16-01868-t003:** Case reports of benign EMT and normal thyroid profile in terms of function, autoimmunity and nodules/cancer in eutopic (cervical) gland (if available); the display starts with the most recent publication date [13,29,74,75,76,77,78,79,80,81,82,83,84,85,86,87,88,89,90,91,92,93,94,95,96,97,98,99,100,101,102,103,104,105,106,107]).

First AuthorReference Number	Publication Year	Sex	Age(y)	Pathological ReportConnective Tissue with Cervical Eutopic Thyroid (If Specified by the Original Authors)	Clinical detectionEndocrine Panel (Thyroid Function)Data on Orthotopic Thyroid (If Specified by the Original Authors)	Imaging Traits of EMTOther Assessments That Were Applied for Diagnosis (If Specified by the Original Authors)	EMT Biopsy or FNA (If Specified by the Original Authors)Surgery for EMT	Outcome
Kolwalkar[13]	2024	M	63	 EMT: colloid adenomatous goiter  no anatomic connection to eutopic gland	 incidental: HRCT of the thorax was done for unrelated fever and cough episode (otherwise asymptomatic)  normal thyroid function on admission  normal eutopic thyroid	 HRCT: EMT of 8.4 × 7.1 × 6.6 cm  CT traits: central necrosis and peripheral calcifications  site: anterior mediastinum  EMT displaced the anterior tracheal wall  posteriorly, EMT had contact with aortic arch, origins of right brachiocephalic trunk, left subclavian and common carotid arteries	 CT-guided Trucut biopsy → thyroid tissue identification (follicular cells and colloid) → excision via midline sternotomy	 no eutopic thyroid removal
Sadidi [74]	2023	F	54	 benign adenoma multinodular atoid goiter  no anatomic connection to eutopic gland	 dyspnea and cough for prior 3 y  normal thyroid function on admission  normal eutopic thyroid (neck ultrasound)	 CT (without contrast): EMT of 5.4 × 5.8 cm  site: anterior mediastinum with extension to middle mediastinum  EMT was on the right side of trachea  intra-operatory: EMT was next to the innominate artery, trachea, and azygos vein	 TTNB→ benign goiter → surgery (right posterolateral thoracotomy incision at IV^th^ intercostal space → EMT resection)	 4-day hospitalization for surgery
Nagireddy[75]	2023	F	57	 double EMTs  colloidal goiter  no anatomic connection to eutopic gland	 1-month history of cough and chest discomfort  normal thyroid function on admission  normal eutopic thyroid	 CT scan: EMTs of 7 × 7 cm + 4.9 × 5 cm  site: upper mediastinum (on both sides)	 CT-guided biopsy (right) indicated EMT → sternotomy	 10-day hospitalization with a good outcome
Melinte[76]	2023	M	40	 benign EMT  no anatomic connection to eutopic gland	 accidental detection of EMT amid CT scan for COVID-19 infection  admission: dysphagia and chest pain (cervical extension)  normal thyroid function  normal thyroid structure	 CT: EMT of 6.1 × 7 × 6.1 cm  site: upper mediastinum  CT trails: calcifications  ^131^iodine scintigraphy: normal cervical thyroid + EMT	 endoscopic ultrasound guided biopsy: EMT (+ecoendoscopy confirmation)  RATS (cervical incision)	
Arjun[77]	2021	F	55	 no malignancy in EMT (situated within the smooth muscle esophageal layer)	 presentation for dyspnea	 radiological evaluation: tumor within the wall of esophagus (firstly, an esophageal leiomyosarcoma was suspected)	 surgical resection → post-operatory confirmation of EMT	
El Haj[78]	2021	F	59	 no malignancy in EMT (32 g)	 4-month history of respiratory complains  normal thyroid function  incidental lung actinomycosis	 CT: EMT of 5.5 × 0.5 × 4.5 cm  site: upper right mediastinum  EMT displaced upper cave vein, and compressed trachea on the contra-lateral site	 bronchoscopy for lung mass→ actinomycosis→antibiotics → lung mass remission → mediastinoscopy → uniportal VATS for EMT resection	 15-day good post-operatory outcome
Sheng[79]	2021	F	48	 benign EMT (post-operatory confirmation)	 admission: 2-month history of dyspnea  normal thyroid function  normal thyroid ultrasound	 echocardiography: intra-cardiac mass → contrast-enhanced ultrasound → CT: mass in right ventricle (5.7 × 4.8 × 4.2 cm)→ CT coronary angiography	 surgery	 16-month follow-up: no recurrence
Imai[80]	2020	F	50	 no malignancy in EMT  no anatomic connection to eutopic gland	 4-month history of cough  normal thyroid function  negative thyroid antibodies against thyroid  normal orthotopic thyroid gland	 CT: EMT of 4 cm  site: upper mediastinum (cranial side; located from thoracic inlet to the superior mediastinum)  CT traits: heterogeneous	 FNA of eutopic and orthotopic thyroid → no malignancy → trans-cervical EMT resection	 good post-operatory outcome
Chen Cardenas[81]	2020case 1	F	49	 no malignancy in EMT	 suspected paraganglioma (otherwise symptoms not connected to EMT)  normal thyroid function and PTH	 CT: 1.2 × 1 × 0.6 cm  site: anterior mediastinum (posterior to manubrium)  ^123^I-MIBG SPECT-CT for suspected paraganglioma (not confirmed)	 EMT resection via mediastinoscopy	
2020case 2	F	42	 no malignancy in EMT	 suspected paraganglioma (otherwise symptoms not connected to EMT)  normal thyroid function and PTH	 CT: 2 × 1.4 × 2 cm  site: anterior left mediastinum  ^123^I-MIBG SPECT- and SPECT-CT for suspected paraganglioma (not confirmed)	 EMT resection via mediastinoscopy	
Kocaman[82]	2020	F	53	 ectopic goiter		 CT: tumor of 8 cm  site: completely intra-pericardial	 thoracoscopy	
Sato[83]	2019	F	69	 EMT: no malignancy	 incidental finding  normal thyroid function	 CT: tumor 2.5 by 2 by 1.5 cm  site: middle mediastinum (completely intra-pericardial)  coronary CT angiography  ^18^F-FDG-PET-CT	 trans-sternal approach for tumor resection	 one year with a good outcome
Carannante[84]	2019	F	30	 hyperplastic cystic nodule in EMT with hemorrhage (of 6.5 × 6 × 2.9 cm; 98 g)	 abdominal pain and vomiting	 CT: EMT of 5.8 × 7.1 cm  site: right side of the mediastinum  CT traits: encapsulated cystic lesion (contact to right upper cave vein and azygos vein, oppositely displaced trachea)  no uptake at PET-CT	 uniportal VATS(mini-thoracotomy at the IV^th^ intercostal space along the right medium axillary line)	 1-month post-operatory good outcome
Regal[85]	2018	F	32	 no malignancy in EMT(10 × 5 cm)  no connection to eutopic thyroid	 incidental detection due to traffic accident  retrospectively: some dyspnea on exertion and heavy exercise  normal thyroid function  normal thyroid at CT and ultrasound	 CT: EMT of 5.2 × 4.4 × 5.2 cm  site: superior upper part of anterior mediastinum (tracheal compression from the left side)  CT traits: heterogeneous structure  fiber optic bronchoscopy: compression of the lateral tracheal wall on the left side (no infiltration)	 midline partial sternotomy	
Ahuja[86]	2018	M	69	 EMT: hyperplastic thyroid nodule  no connection to eutopic thyroid	 admission: weight loss, left side flank pain + 5-y history of periodic chest pain  5 year prior: nephrectomy for renal oncocytoma  normal thyroid function  normal eutopic tissue	 CT:EMT of 4.5 × 3.3 × 2.7 cm  site: near left atrium  CT traits: heterogeneous  I^123^-MIBG + SPECT excluded a paraganglioma	 sternotomy	 normal post-operatory thyroid function
Raji case 1[87]	2018	F	“elderly”	 EMT: nodular hyperplasia  no connection to eutopic thyroid	 recent dyspnea, headache, and somnolence	 CT angiography with contrast: EMT  site: anterior mediastinum  CT traits: heterogeneous  FDG PET-CT confirmation	 partial upper sternotomy + anterior mediastinal mass resection	 4-month post-operatory good outcome
Metere[88]	2018	M	63	 multinodular goiter features in EMT  thin strip of tissue between left thyroid lobe and EMT at contrast-enhanced CT	 asymptomatic  normal thyroid function  normal thyroid antibodies	 CT: 6 × 8 cm  site: para-tracheal mass laying on the right bronchus  CT traits: heterogeneous	 transverse cervicotomy for left thyroid lobe → a longitudinal sternal splitting (upper partial sternal split, extended to the 4th intercostal space) to remove the mediastinal mass→ complete the thyroidectomy	
Robitaille[89]	2017	F	58	 being EMT at cytological report	 incidental detection during investigations (CT) for prior ovarian cancer (treated with chemotherapy and surgery)	 CT: 1.2 cm  CT: right para-tracheal tumor  ^18^F-FDG/PET: no uptake	 EBUS-TBNA: EMT confirmation (no removal)	 follow-up amid ovarian cancer protocol
Tan[90]	2017	F	53	 benign EMT	 2-y history of chest pain  normal thyroid function	 transthoracic & transesophageal echocardiography firstly identified the tumor in right cardiac ventricle (4.5 × 4.1 cm)  coronary angiography CT: feeding vessel was a branch of the left anterior descending artery	 surgical excision	 10-day hospitalization
Hardy[91]	2016	M	35	 benign EMT	 presentation with weight loss, cough and breathlessness  normal thyroid function  negative TPOAb	 99mTc scintigraphy and PET-CT showed no uptake (non-diagnostic)	 EBUS-TBNA: established the diagnostic of EMT	
Abdel Aal[92]	2015	F	77	 no malignancy in EMT (histological report through biopsy)  no connection to eutopic gland	 incidental detection amid breast cancer protocol (at CT scan)  normal thyroid function	 CT: at first, EMT was suspected to be a breast cancer metastasis  EMT of 7 × 5.8 × 5 cm  EMT site: anterior mediastinum (on the left side of the aortic arch; right tracheal deviation)  99m-Tc-HDP bone scan: no metastases	 CT-guided percutaneous transthoracic punch (core) biopsy: showed EMT, not breast cancer metastasis	 breast cancer protocol
Wang [93]	2014	F	45	 double ectopic thyroid: EMT and lateral cervical  no connection to eutopic gland	 10-day history of chest pain  normal thyroid function  negative antibodies against thyroid	 CT: neck ectopic thyroid of 2.4 × 1.4 × 2.8 cm and EMT of 4 × 2.5 cm  site: EMT in anterior mediastinum  99m-Tc scintigraphy for EMT	 ectopic tissues were removed via VATS in addition to using a neck incision	 no removal of eutopic gland
Scognamillo[94]	2014case 1	F	55	 benign EMT	 dyspnea + dysphagia  normal thyroid function	 anterior mediastinum	 total thyroidectomy to allow EMT removal without sternotomy	 good post-operatory outcome
2014case 2	F	56	 benign EMT	 dyspnea  normal thyroid function
Roh [95]	2013	M	65	 benign EMT (upon biopsy)	 incidental detection amid CT upon a traffic accident (asymptomatic)  normal thyroid function  negative anti-thyroid antibodies	 CT: EMT of 4.5 × 2.9 cm  site: right para-tracheal mass (upper cave vein compression, but not symptoms)  CT traits: heterogeneous EMT	 EBUS-TBNA: normal thyroid follicles in EMT	 no removal (proposed surgery)
Waltz[96]	2013	M	31	 follicular thyroid parenchyma (6.2 g)  no connection to eutopic gland	 dysphagia  sensation of sternal pressure	 EMT site: upper anterior mediastinum  blood supply from internal mammary artery (intra-operatory)	 biopsy by CT-guided FNA  excision via cervical incision:removal of the left thyroid lobe to allow access to EMT excision	 rapid post-operatory recovery
Thuillier[97]	2012	F	77	 benign EMT	 1-month history of dyspnea  normal thyroid function  normal eutopic thyroid	 EMT site: anterior mediastinum		
Mace[98]	2011	F	80	 benign EMT  no connection to eutopic gland	 incidental detection amid MRI  normal thyroid function	 EMT site: anterior mediastinum (tracheal deviation)—of 5 cm	 ultrasound-guided FNA: no malignancy (cytological report) → surgery via cervical incision: removal of right thyroid lobe followed by EMT excision	 rapid post-operatory recovery
Kumaresan[99]	2011	F	30	 benign EMT	 radiating neck pain  normal thyroid function  normal thyroid appearance	 MRI: para-tracheal mass  Tc99m pertechnetate scintigraphy: hyper-function in EMT and suppression of eutopic gland activity	 CT-guided FNA cytology: thyroid adenoma  surgical excision	 Tc99m pertechnetate scintigraphy at 6 weeks since EMT removal: normal eutopic gland
Pilavaki [100]	2009	M	72	 EMT; multinodular goiter  no connection to eutopic gland	 2-month history of cough	 CT  site: right side of the mediastinum  MRI: multi-cystic EMT	 excision via right lateral thoracotomy(diameter of 7.5 cm; blood supply: from intrathoracic vessels)	 1-y post-operatory good outcome
Topcu[101]	2009	M	68	 benign EMT  no connection to eutopic gland	 EMT + bronchial anomalies in a patient with newly detected lung cancer  normal thyroid function	 CT: EMT of 6 × 4 × 4 cm  bronchoscopy  PET-CT: increased uptake in lung mass + left thyroid lobe, but not in EMT  ^131^iodine scintigraphy: EMT	 CT-guided trans-thoracic FNA showed non–small cell lung carcinoma  posterolateral thoracotomy for lung cancer + EMT	
Guimarãescase 1[102]	2009	F	40	 EMT: 12 × 9 cm (400 g): nodular thyroid hyperplasi + micro-follicles containing colloid + inflammatory reaction	 3-week history of dyspnea, cough, night sweats, asthenia, myalgia, nausea and subfebrile temperatures  normal thyroid function	 CT scan was done after 8 days of antibiotics for supposed bacterial pneumonia  CT: EMT of 10 cm  EMT site: anterior mediastinum  additional MRI and 99m-Tc scintigraphy	 attempt to reduce EMT by levothyroxine suppression therapy  surgical excision + total thyroidectomy	
Karapolat[103]	2009	M	74	 benign EMT (follicular patern)	 chest pain  normal thyroid function	 CT: EMT of 5 × 5 cm  site: right posterior mediastinum  bronchoscopy: not helpful for diagnosis	 trans-bronchial FNA biopsy: not successful  right thoracotomy → biospy with EMT confirmation → EMT resection	 1-y good post-operatory outcome
Chataigner[29]	2007	F	64	 benign EMT	 incidental detection  prior treated breast cancer (surgery and radiotherapy)	 transthoracic and transesophageal echocardiography detected the tumor (of 3 × 3 cm)  CT: EMT of 5.5 × 4 cm  site: extra-cardiac, extra-pulmonary, intra-pericardial EMT (adjacent to ascending aorta)	 median sternotomy with complete EMT excision	 2-y post-operatory asymptomatic outcome
Sakorafas[104]	2004	NA	NA	 ectopic intrathoracic thyroid			 right lateral thoracotomy	
Van Schil[105]	1989	M	56	 EMT: nodular hyperplasia + focal lymphocytic thyroiditis		 CT: para-tracheal mass	 surgery	
Arriaga[106]	1988	M	46	 EMT: colloid follicles + minimal nodular changes	 extrinsic compression of the esophagus  normal thyroid function	 CT: retro-esophageal mass of 5 cm  EMT site: upper mediastinum  99m-Tc pertechnetat scintigraphy	 surgery (left cervical incision)	
Asp[107]	1987	NA	NA			 thallium 201/99m-Tc pertechnetate radionuclide study for localization: EMT was mistaken as parathyroid adenoma		

Abbreviations: cm = centimeter; CT = computed tomography; EBUS-TBFNA endobronchial ultrasound-guided transbronchial needle aspiration; EMT = ectopic mediastinal thyroid; FDG PET-CT = fludeoxyglucose positronic emission tomography-CT; FNA = fine needle aspiration; F = female; HRCT = High Resolution CT; MRI = magnetic resonance imagery; ^123^I-MIBG = iodine meta-iodobenxylguanidine; M = male; NA = not available; PTH = parathormone; RATS = robotic assisted thoracoscopic surgery; SPECT = single photon emission computed tomography; Tc = technetium; TTNB = transthoracic needle biopsy; VATS = video-assisted thoracic surgery; y = year; green font = any data on thyroid profile or orthotopic gland.

**Table 4 cancers-16-01868-t004:** Case reports of benign EMT and prior or concurrent thyroid diseases; the display starts with the most recent publication date [11,22,34,35,73,87,102,115,116,117,118,119,120,121,122,123,124,125,126,127,128,129,130].

First AuthorReference Number	Publication Year	Sex	Age(y)	Pathological ReportConnective Tissue With Cervical Eutopic Thyroid (If Specified by the Original Authors)	Clinical DetectionEndocrine Panel (Thyroid Function)Data on Orthotopic Thyroid (If Specified by the Original Authors)	Imaging traits of EMTOther Assessments That Were Applied for Diagnosis (If Specified by the Original Authors)	EMT Biopsy or FNA (If Specified by the Original Authors)Surgery for EMT	Outcome
Khan[22]	2021	F	19	 double EMTs (lumbar and posterior mediastinal)  no malignancy in EMT	 accidental detection of a large thyroid nodule on left lobe (3.3 × 3.7 × 3.6 cm) → left hemi-thyroidectomy → detection of a lumbar lump (that was considered a lipoma)→ resection of lumbar lump→ confirmation of ectopic thyroid → total thyroidectomy	 chest MRI detection of a posterior EMT (1.1 × 0.7 cm) (**)	 EMT resection via thoracotomy	 post-operatory radioiodine ablative therapy  TSH suppression therapy (**)
Tsai[11]	2021	F	50	 no malignancy in EMT	 incidental finding of EMT and thyroid nodule amid CT + PET-CT for prior pulmonary adenocarcinoma  FNA for thyroid nodule: papillary carcinoma	 CT + PET: EMT of 2.9 × 1.6 cm	 EBUS-TBFNA: benign EMT	
Kola[73]	2021	M	42	 no malignancy in EMT	 3-month history of dyspnea, chest pain, fatigue  TSH = 0.33 (normal: 0.35–4.94) mU/L  incidental thyroid nodule (left lobe) of 0.6 cm  incidental adrenal tumor (right gland) of 5 × 3 cm	 CT: EMT of 9 × 6 cm  99m-Tc pertechnetate scintigraphy for eutopic thyroid (no data on EMT): low uptake of 0.2% (normal: 0.35–3.65)—probably a prior thyroiditis	 surgical removal (no other data) → post-operatory confirmation of EMT	 1-month after surgery: good outcome (normal TSH, T3, T4)
Muzurović[115]	2021	F	53	 colloid goiter with cystic transformation in giant EMT  no malignancy in eutopic thyroid  two parathyroid adenoma  no anatomic connection to eutopic gland	 3-month history of cough, and chest pain  normal thyroid function  negative thyroid antibodies against thyroid  incidental thyroid dominant nodule (left lobe)  biological confirmation of primary hyperparathyroidism  incidental two parathyroid tumors: right inferior + ectopic mediastinal	 CT: EMT of 9.5 × 7.5 × 11.5 cm  site: upper anterior mediastinum  CT traits: heterogeneous, central calcifications  EMT displaced thoracic aorta, pulmonary artery and trachea  99m-Tc pertechnetate scintigraphy for eutopic + ortotopic thyroid  99m-Tc sestamibi scintigraphy with SPECT-CT for both parathyroid tumors	 CT-guided biopsy: EMT confirmation  partial sternotomy (resection of EMT + total thyroidectomy + selective parathyroidectomy + thymectomy via mediastinal and cervical exploration and)	 normal post-operatory PTH  and thyroid function  3-month post-operatory good outcome
Rajaraman[116]	2019	NA	NA		 thyrotoxicosis with hyper functioning thyroid gland	 two abnormal foci of uptake in the mediastinum at SPECT		
Agrawal[34]	2019	M	41	 no histological or cytological confirmation (only imaging aspects)  no connection to orthotopic thyroid at scintigraphy	 admission for recent onset of Graves’s disease  hyperthyroidism  atoimmune thyroiditis (Graves’s disease)  thyroid ultrasound: enlargement according to Graves’ disease	 Tc-99m sodium pertechnetate scintigraphy: diffuse goiter + suspected EMT  SPECT-CT (low-dose, non-contrast CT) EMT: on the right side within the upper mediastinum (of 1 × 1 cm), posterior to manubrium	 no EMT biopsy  no EMT surgery	 medical Therapy for Graves’s disease
Sohail[117]	2019case 1	F	68	 benign nodular hyperplasia with degenerative changes in both tissues changes  separated capsule of EMT	 high blood pressure  progressive anterior neck swelling (eutopic goiter)  3-month history of dyspnea, dysphagia  dilated neck veins  normal thyroid function  large multinodular goiter	 CT: multinodular goiter with multiple calcifications + retrosternal extension in anterior mediastinum	 total thyroidectomy + sternotomy  (intra-operatory finding: EMT)	 respiratory distress on 2nd post-op day (iatrogenic right phrenic nerve injury)
2019 case 2	F	42	 multinodular eutopic goiter and EMT (with distinct capsule, of 5 × 5 cm)	 anterior neck swelling (eutopic goiter) for the last 5 years (+progressive dyspnea)	 CT: multinodular goiter with right lobe and isthmus extension into superior mediastinum (of 8.8 × 6.5 × 4.5 cm)	 thyroid FNA: benign  total thyroidectomy + sternotomy, but initial transverse neck incision (intra-operatory finding: EMT)	 3-day hospitalization  6-week post-operatory good outcome
Rajicase 2[87]	2018	M	“elderly”	 colloidal thyroid tissue  no connection to eutopic thyroid	 right hilar mediastinal mass on chest radiograph  thyroid nodule at eutopic gland	 CT: right para-tracheal mass  F-18 FDG PET-CT: confirmation  99-Tc-pertechnetate: normal uptake and no EMT confirmation  FNA of the thyroid nodule (benign)	 FNA at eutopic thyroid: benign  mediastinoscopy with biopsy of EMT  right sided VATS	 1-y follow-up: good outcome
Hummel[118]	2017	F	61	 benign EMT	 presentation as emergency for cough and right chest pain (firstly a pulmonary embolism was excluded via CT)  prior history of total thyroidectomy for thyroiditis and nodules and consecutive iatrogenic hypothyroidism	 SPECT-CT:2 × 1.7 cm  site: right para-tracheal EMT  ^123^iodine radioiodine uptake + SPECT-CT	 EBUS-guided biopsy followed by I-123 SPECT-CT	 conservative approach  1-y stationary EMT
Wang[119]	2017	F	53	 benign EMT	 2-hystory of intermittent precordial pain  transitory hypothyroidism for 3 months following EMT removal	 intra-cardiac EMT (right ventricle)  CT: EMT of 4.8 × 4.1 cm  plain + contrast enhanced CT showed EMT and thyroid with the same traits	 surgery (thoracotomy)	 3-month post-operatory hypothyroidsim
Patel[120]	2016	F	54	 benign EMT	 incidental detection of a tracheal deviation at X-ray  prior history of total thyroidectomy for Hashimoto’s thyroiditis and diffuse cervical goiter and consecutive iatrogenic hypothyroidism	 CT: 6.4 × 3.2 cm  site: upper mediastinum on the left side (tracheal deviation)	 surgical resection via trans-cervical approach (of note, prior thyroidectomy was done 5 y before)	
Serim[121]	2016	F	62	 benign hyper-functional EMT	 2-y history of hyperthyroidism  hyperthyroidism on admission:  FT3 = 4.5 ng/mL (1.8–5)  FT4 = 1.83 ng/mL (0.8–1.9)  TSH = 0.016 µU/mL (0.4–4)	 99m-Tc pertechnetat scintigraphy: increased uptake in EMT and normal in eutopic thyroid → CT: EMT of 6 × 6 × 4.5 cm  EMT site: left superior anterior mediastinum (no tracheal or esophageal effects)  CT traits: heterogeneous and calcifications	 therapy with propil thyiouracil → total thyroidectomy + EMT excision	
Cunha[122]	2016	F	67	 benign hyper-functional EMT	 admission for thyrotoxicosis persistent after levothyroxine replacement was stopped  prior history of total thyroidectomy for non-toxic goiter (7 y ago) and consecutive iatrogenic hypothyroidism  hyperthyroidism on admission:  high thyroglobulin: 294 mg/mL  positive TRAb = 19.7 U/L	 whole body scintigraphy with ^131^iodine with SPECT-CT showed EMT  CT: EMT of 6 × 4 cm	 levothyroxine stopped → methimazol for 3 months → normal thyroid function → EMT surgical resection (via horizontal cervical incision)	
Kesici[35]	2015	F	49	 EMT: nodular hyperplasia + chronic thyroiditis (no malignancy)  papillary thyroid carcinoma in eutopic thyroid	 detection via thyroid scintigraphy after total thyroidectomy for multinodular goiter underlying a papillary thyroid carcinoma  a 3–4 y history of swelling (no related to eutopic thyroid, but with EMT)	 CT: EMT of 3 × 3 cm	 median sternotomy	 radioiodine therapy (100 mCi) after EMT removal
Kamaleshwaran[123]	2015	F	29	 EMT: benign	 detection via thyroid 131 iodine scintigraphy after total thyroidectomy for multinodular goiter underlying a papillary thyroid carcinoma  eutopic papillary thyroid carcinoma	 SPECT-CT: EMT of 7 × 6 × 4 cm (with calcification)	 mediastinal mass excision through sternotomy + thymectomy (prior total thyroidectomy)	 radioiodine therapy (100 mCi) after EMT removal → after 6 months: undetectable thyroglobulin + negative iodine 131 whole body scintigraphy
Lee [124]	2014	F	62	 benign EMT	 atypical chest pain + dysphagia  situs inversus  prior history of left hemi-thyroidectomy for a thyroid nodule with hyperthyroidism  normal thyroid function	 CT: EMT of 7.3 × 5.3 × 3.5 cm  EMT site: posterior mediastinum  iodine scintigraphy: uptake within right lobe and reduced uptake in EMT	 EMT resection via thoracotomy (left posterolateral incision )	 10-day hospitalization amid EMT removal  good post-operatory outcome
Barker[125]	2012	F	37	 EMT: multinodular goiter  no connection to eutopic gland	 recurrent choroidal inflammation, night sweats + dyspnea  subclinical thyrotoxicosis:  FT4 = 16.6 pmol/L(12–22)  TSH = 0.11 mUI/L(0.3–4.6)  multinodular eutopic goiter	 CT: intra-thymus EMT  EMT site: anterior mediastinum	 biopsy via rigid bronchoscopy + left VATS: normal thymus tissue → de novo left breast ductal and lobular carcinoma) →median sternotomy + radical excision of EMT for diagnosis	 mastectomy was done after 3 months since EMT removal (since EMT was first suspected to be a metastasis)
Demirhan[126]	2009	M	62	 benign EMT	 admission as emergency for atypical chest pain + dysphagia (for 3 months)  clinically normal thyroid function  serum “mild TSH elevation” with normal FT3 and FT4	 posterior mediastinum (esophageal compression)	 resection via sternotomy (7 × 5 × 5 cm; blood supply: from the innominate artery + left innominate vein)	 Good post-operatory outcome for 9 months of follow-up
Guimarãescase2[102]	2009	F	57	 EMT: benign macro-follicular pattern	 dyspnea and asthenia  history of right partial thyroidectomy (9 y before) for nodular hyperplasia → hypothyroidism → T4 therapy → normal thyroid function  negative antibodies against thyroid	 CT: EMT of 4.5 × 4.3 cm  EMT site: anterior, upper mediastinum  additional 99m-Tc scintigraphy	 surgical excision	
Bodner [127,128]	2005	F	72	 benign EMT (“thyroid adenoma”)	 synchronous large goiter that required surgery	 EMT was surrounded by azygos vein, upper cave vein and confluence of the innominate vein	 thoracoscopic resection with da Vinci robot (first use)	
Gamblin[129]	2003	F	62	 benign EMT  no connection to the eutopic gland	 dyspnea, cough  history of hyperthyroidism + multinodular goiter → current medication with anti-thyroid drugs (tapazol)	 CT site: anterior mediastinum (compression of the left innominate vein; blood supply from the thoracic vessels)	 left hemi-thyroidectomy + isthmectomy + transternal approach (median sternotomy)	
Basaria[130]	1999	NA	NA	 EMT: potential pathogenic role of TRAb	 substernal chest pain  prior history of Graves’ disease (9 y before)		 surgically removed	

Abbreviations: cm = centimeter; CT = computed tomography; EBUS-TBFNA endobronchial ultrasound-guided trans-bronchial needle aspiration; EMT = ectopic mediastinal thyroid; FDG PET-CT = fludeoxyglucose positronic emission tomography-CT; FNA = fine needle aspiration; F = female; FT3 = free triiodothyronine; FT4 = free thyroxine; MRI = magnetic resonance imagery; M = male; NA = not available; PTH = parathormone; SPECT = single photon emission computed tomography; Tc = technetium; TSH = Thyroid Stimulating Hormone; T3 = triiodothyronine; T4 = thyroxine; TRAb = TSH receptor antibodies; y = year; (**) no specification of CT scan; neither the confirmation of the type of malignancy within the eutopic thyroid.

**Table 5 cancers-16-01868-t005:** The clinical elements or scenario of detection in EMTs according to our methods [7,9,11,13,22,29,34,35,36,37,41,42,45,48,49,50,51,52,53,54,58,59,60,61,62,63,64,65,66,67,68,69,70,71,72,73,74,75,76,77,78,79,80,81,82,83,84,85,86,87,88,89,90,91,92,93,94,95,96,97,98,99,100,101,102,103,104,105,106,107,115,116,117,118,119,120,121,122,123,124,125,126,127,128,129,130] (blue font = incidentaloma).

Complain (Sign or Symptom) or Scenario of EMT Detection	Original Reference Number
sternal (chest) pressure/pain	[9,73,75,76,86,90,93,96,103,115,126,130]
cough	[7,9,13,50,55,69,74,75,80,91,100,102,115,129]
dysphagia	[9,66,76,94,96,117,126]
dyspnea	[9,50,63,69,73,74,77,79,85,87,91,94,97,102,117,125,129]
precordial pain	[119]
radiating neck pain	[99]
compressive symptoms	[48]
intermittent night sweats	[9,102,125]
hemoptysis	[45]
mediastinal syndrome	[52,60]
rash at sternal level	[37]
spine compression (vertebral metastasis)	[58]
weight loss	[37,86,91]
clavicular metastasis	[61]
upper vein cave syndrome	[55]
hoarseness	[62]
hemi-thorax pain	[64]
unrelated fever	[13]
abdominal pain and vomiting	[84]
left side flank pain	[86]
headache	[87]
somnolence	[87]
asthenia	[102]
fatigue	[73]
myalgia	[102]
sub-febrile temperature	[102]
dilated neck veins	[117]
emergency mimicking a pulmonary embolism	[118]
clinical picture of thyrotoxicosis	[121,122]
incidental amid MRI or CT	[7,11,29,52,59,70,76,83,85,89,92,95,98,120]

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
