# Peer review of "Personalized Management of Malignant and Non-Malignant Ectopic Mediastinal Thyroid: A Proposed 10-Item Algorithm Approach"

_cancers, 2024, doi:10.3390/cancers16101868_

Round 1

Reviewer 1 Report

Comments and Suggestions for Authors

Dear authors,

I have now completed the review of the manuscript titled "Personalized management of malignant and non-malignant ectopic mediastinal thyroid; a proposed 10-item algorithm of approach."

In this comprehensive review on ectopic mediastinal thyroid (EMT), the authors have done an impressive job analyzing a large body of literature on this rare condition.

The manuscript is interesting and, in general, fairly well-written.

I have some suggestions to further improve the quality of the manuscript.

I would like to suggest that the authors address these limitations in the article, either by discussing them in the limitations section or, where feasible, by making the appropriate revisions:

1. The review covers a lot of ground, from clinical presentation to imaging to management of EMT. While thorough, the broad scope makes it a bit unfocused at times. Tightening the narrative around a few key themes or questions could improve readability.

2. One case had history of atopic skin disease in addition to ectopic mediastinal thyroid. Some article related to Association between atopic dermatitis and school readiness in preschool children, which explores associations between atopic dermatitis and child development, may be relevant if studying comorbidities in patients with ectopic thyroid. Consider discussing it. Also, one patient found to have ectopic mediastinal thyroid, understanding environmental factors associated with COVID-19 incidence may provide useful epidemiological background information. I was able to find some articles like Effects of meteorological factors and air pollutants on the incidence of COVID-19 in South Korea, and Retinal Vascular Occlusion Risks during the COVID-19 Pandemic and after SARS-CoV-2 Infection, but was unable to find more due to time limitation. Authors should find more by themselves.

3. The analysis of malignancy rates in EMT is interesting, with the 18.8% incidence being higher than suggested by prior reports. Related research on this point would be prudent. Therefore I would like to suggest expansion of discussion section, including global burden of disease study. For example, recent Global, regional, and national incidence of six major immune-mediated inflammatory diseases: findings from the Global Burden of Disease Study 2019 can be used discussing autoimmune thyroiditis in some patients with ectopic mediastinal thyroid.

4. The 10-item algorithm proposed for EMT management is a helpful synthesis of the findings. However, it seems to be based more on the authors' interpretation of the literature rather than rigorous evidence or consensus guidelines. Framing it more explicitly as an expert opinion piece would avoid overstating the conclusiveness of the recommendations.

Thank you for your valuable contributions to our field of research. I look forward to receiving the revised manuscript.

Author Response

Response to Review 1 Comments

Dear Reviewer,

Thank you very much for your time and your effort to review our manuscript.

We are very grateful for providing your valuable feedback on the article.

Here is our response and related amendment that has been made in the manuscript according to your review (marked in yellow color).

Dear authors,

I have now completed the review of the manuscript titled "Personalized management of malignant and non-malignant ectopic mediastinal thyroid; a proposed 10-item algorithm of approach."

  1. In this comprehensive review on ectopic mediastinal thyroid (EMT), the authors have done an impressive job analyzing a large body of literature on this rare condition.

Thank you very much. We really appreciate it!

  1. The manuscript is interesting and, in general, fairly well-written.

I have some suggestions to further improve the quality of the manuscript.

Thank you very much. We followed your recommendations. Thank you.

  1. I would like to suggest that the authors address these limitations in the article, either by discussing them in the limitations section or, where feasible, by making the appropriate revisions:

Thank you very much. We expanded the Limitations section according to your suggestions and observations as shown below. Thank you

  1. The review covers a lot of ground, from clinical presentation to imaging to management of EMT. While thorough, the broad scope makes it a bit unfocused at times. Tightening the narrative around a few key themes or questions could improve readability.

Thank you very much. We introduced a Synopsis with the key themes and the main associated tables that we presented in order to highlight each topic in each sub-section at Results. Of note, the spectrum of aspects is extremely complex and a massive work has been done in this particular matter, hence, the synthesis is intended to offer a glimpse perspective. Thank you

  1. One case had history of atopic skin disease in addition to ectopic mediastinal thyroid. Some article related to Association between atopic dermatitis and school readiness in preschool children, which explores associations between atopic dermatitis and child development, may be relevant if studying comorbidities in patients with ectopic thyroid. Consider discussing it. Also, one patient found to have ectopic mediastinal thyroid, understanding environmental factors associated with COVID-19 incidence may provide useful epidemiological background information. I was able to find some articles like Effects of meteorological factors and air pollutants on the incidence of COVID-19 in South Korea, and Retinal Vascular Occlusion Risks during the COVID-19 Pandemic and after SARS-CoV-2 Infection, but was unable to find more due to time limitation. Authors should find more by themselves.

Thank you very much. This is part of the large chapter with respect to the endocrine disruptors and developmental issues in the field of endocrine glands, including ectopic thyroid. Thank you for pointing this aspect. According to your interesting suggestions, we added at Limitations section the followings: “As mentioned, a single pediatric case was found [68], but, generally, the connection between the endocrine disruptors in terms of environmental factors (such as the geographic influence, the air pollutions, etc.) or internal factors like infectious conditions (as recently shown by the COVID-19 pandemic) or autoimmune (namely, endocrine and non-endocrine autoimmunity) issues should be studied in relationship with the developmental conditions, specifically, the ectopic thyroid (either acting during pregnancy or early during childhood); yet, this represents a future topic to be studied [235-240].” Thus, another five references have been added (from number 135 to number 140).Thank you

  1. The analysis of malignancy rates in EMT is interesting, with the 18.8% incidence being higher than suggested by prior reports. Related research on this point would be prudent. Therefore I would like to suggest expansion of discussion section, including global burden of disease study. For example, recent Global, regional, and national incidence of six major immune-mediated inflammatory diseases: findings from the Global Burden of Disease Study 2019 can be used discussing autoimmune thyroiditis in some patients with ectopic mediastinal thyroid.

Thank you very much for pointing the disease burden in the thyroid domain. We added also as a future topic of study with regard to this aspect as following: “Further on, which the exact impact on the overall thyroid cancer-associated burden with respect to the ectopic thyroid-related malignancy represents an open issue according to the current level of statistical evidence; we expect a future expansion amid increased awareness and access to advanced imagery tools and performant surgical access with an overall good long term outcome in ectopic thyroid cases [241]. (GBD 2019 IMID Collaborators. Global, regional, and national incidence of six major immune-mediated inflammatory diseases: findings from the global burden of disease study 2019. EClinicalMedicine. 2023:64:102193. doi:10.1016/j.eclinm.2023.102193.)

  1. The 10-item algorithm proposed for EMT management is a helpful synthesis of the findings. However, it seems to be based more on the authors' interpretation of the literature rather than rigorous evidence or consensus guidelines. Framing it more explicitly as an expert opinion piece would avoid overstating the conclusiveness of the recommendations.

Thank you very much. We followed your recommendation and added to the 10-item algorithm (now, this is the section number 2.11.) the followings: “of note, this is our interpretation and opinion based on the current literature evidence, not a consensus guideline”. Thank you

  1. Thank you for your valuable contributions to our field of research. I look forward to receiving the revised manuscript.

Thank you very much. We really appreciate it.

Thank you very much.

Reviewer 2 Report

Comments and Suggestions for Authors

The manuscript “Personalized management of malignant and non-malignant ectopic mediastinal thyroid; a proposed 10-item algorithm of approach," cancers-2975667, reviews published EMT studies and proposes an algorithm that could be useful for EMT identification and EMT patients’ management. The first part of the paper, particularly the introduction section, is written very well; it is easy-to-follow and contains some interesting observations and summaries. But the rest of the paper is very hard to follow; the tables are too large and confusing. Furthermore, the main result of the paper (the proposed algorithm) is presented in the discussion section. Therefore, as long as the paper contains some useful information about EMT, it needs some major corrections (before all, in the means of presentation and text reorganization) before being reconsidered for publication.

1. In general, the paper should be properly shortened and restructured. Written in this way, it lacks focus, the data are confusedly presented, the authors fade away in the descriptions of individual previously published papers, and a significant portion of the publication is spent to simply listing the data that has already been published. This should be a review paper, and in the review paper, all previously published data should be summarized, compared, and possible differences discussed. That is a key part of any review paper, and that is missing here.

2. A simple summary lacks information about your main result (the proposed algorithm). Written in this way, it totally differs from the title of the manuscript. Please make suitable corrections and adjust the text.

3. The abstract should be shortened (sufficient data should be removed) and rewritten in a more focused way.

4. Figure 1 – shows that 218 full text publications were reviewed for eligibility, 70 were removed, and 89 were included in the final study. But 218-70=148. An explanation is required here.

5. All abbreviations should be placed under the table.

6. Improve table presentation for better readability and brevity. I recommend that you remove tables from the body text of the manuscript and provide them as a supplement. You may also present them graphically.

7. The authors only listed previously published data (tables 1 and 2). Please make comparisons between the tables, draw conclusions, and explain them thoroughly.

8. Section 3.7 with Fig. 6 must be shifted into the results section. This is your main result. Data given in its original order renders Fig. 2 (and associated lines 714-717) confusing since you refer to "our analysis" without documenting its previous performance or presentation.

9. On the other hand, some parts from the results section should be shifted into the discussion section, as they are discussing previously published data.

10. The pies should be presented in a uniform way.

11. Please clarify and explain Figure 4— presentation, findings, and conclusions.

12. The list of references should be shortened.

Author Response

Response to Review 2 Comments

Dear Reviewer,

Thank you very much for your time and your effort to review our manuscript.

We are very grateful for your insightful comments and observations, also, for providing your valuable feedback on the article.

Here is a point-by-point response and related amendments that have been made in the manuscript according to your review (marked in yellow color).

  1. The manuscript “Personalized management of malignant and non-malignant ectopic mediastinal thyroid; a proposed 10-item algorithm of approach," cancers-2975667, reviews published EMT studies and proposes an algorithm that could be useful for EMT identification and EMT patients’ management. The first part of the paper, particularly the introduction section, is written very well; it is easy-to-follow and contains some interesting observations and summaries.

Thank you very much. We really appreciate it.

  1. But the rest of the paper is very hard to follow; the tables are too large and confusing. Furthermore, the main result of the paper (the proposed algorithm) is presented in the discussion section. Therefore, as long as the paper contains some useful information about EMT, it needs some major corrections (before all, in the means of presentation and text reorganization) before being reconsidered for publication.

Thank you very much. We revised the data presentation and also moved the 10-item algorithm to Results (section 2.11).

Of note, this is to our aware the most complex work in the field of EMT so far and we had to cover a massive, multidisciplinary panel across of more than 3 decades of publications. The data from literature were not homogenous, the scenario of detection, management and outcome might be completely distinct and unusual, and that is why a difficult selection of the cases we particularly introduced was done and seemed extremely important for the general perspective and according to the current understanding and current level of statistical evidence.

Moreover, each subsection has a brief conclusion for each specific major segment to make it easier for readers amid a multidisciplinary perspective. For example:

Section 2.1. (studies in EMT)

“To summarize, these are all retrospective studies (n = 10; N = 36 subjects with EMT, and one them had a papillary thyroid carcinoma in EMT [52]) with various endpoints [7,9,45,48-54] such as:

  • Standford database (N = 7 patients with benign EMTs) was analyzed across two distinct papers, but this was the same cohort [9,48];
  • a single-center study on surgical outcome (between 1991 and 2006) amid approaching cervico-mediastinal goiters (N = 97 individuals) identified 11 of them had as having a “forgotten” goiter in mediastinum (an alternative name to EMT) [51];
  • a study on 3092 patients who underwent thyroidectomy (between 2000 and 2013) identified 28/3092 of them with ectopic thyroid tissue of any type; among this subgroup, 5 out of 28 had EMT (female to male ratio of 3 to 2; mean age of 41 years) [52];
  • a single-center study in ectopic thyroid tissue (N = 47) of any type (between 2013 and 2022) identified 4 out of these 47 individuals with benign EMT (female to male ratio of 4 to 1, mean age of 55.5 years; of note, one subject had a synchronous thymus lipoma) [7];
  • a case series on quality assurance protocol in endobronchial ultrasound-guided transbronchial needle aspiration (EBUS-TBNA) included 3 EMT cases (female to male ratio of 2 to 1; average age of 84.33 years) [49];
  • a case series of benign EMT (N = 3 subjects; female to male ratio of 2 to 1, mean age of 52.33 years) focused on using endobronchial ultrasound-guided fine needle aspiration (EBUS-FNA) [50];
  • a 8-patient series with huge mediastinal masses identified one benign EMT in a 44-year-old male [45];
  • a retrospective study in 665 patients who underwent thyroidectomy (between 2005 and 2012) identified one subject with EMT [53];
  • a study on 16 patients with benign ectopic aberrant thyroid identified one patient with EMT [54];”

Section 2.2. (malignancy presentation in EMT):

“To summarize, the histological types of the primary thyroid malignancies in 22 subjects (Table 1 and 2) confirmed with EMT-related cancer were as follows:

  • papillary (the most common type; the patients had any form from micro-carcinoma to severe metastatic disease); of note, one more case was introduced in prior studies-based analysis, hence, a total of eleven subjects [52]
  • follicular variant of the papillary type (N = 2)
  • Hürthle-cell thyroid follicular malignancy (N = 1)
  • poorly differentiated (N = 1)
  • anaplastic (N = 2)
  • medullary (N = 1)
  • lymphoma (N = 2)
  • MALT (mucosa-associated lymphoid tissue) (N = 1) [36,37,41,42,55,58-67,69-72].
  • an additional case of immature teratoma does not belong to the specific category of primary thyroid cancer, but it was identified in ectopic tissue (EMT) [68]”

Section 2.3. (all data in EMT):

“To summarize, a total of 117 patients with any type of EMT were described across the papers we identified according to our methods, namely:

  • 36 subjects (and one of them with malignant EMT [52]) were confirmed in studies with various endpoints (other than specifically evaluating the EMT population) or case series of at least three EMT patients per series [7,9,45,48-54]
  • 21 subjects diagnosed with any type of malignancy in EMT [36,37,41,42,55,58-72] plus the mentioned case above [52] (N = 22 persons with malignant EMT)
  • 37 subjects with benign EMT and otherwise normal thyroid profile [13,29,74-107]
  • 23 subjects with benign EMT and a secondary thyroid condition of any type [11,22,34,35,73,87,102,115-130]”

Section 2.3.A. (thyroid anomalies in EMT):

“Our sample-based analysis (Tables 1,2,3,4) showed hypothyroidism as following:

  • congenital type [52]
  • iatrogenic hypothyroidism (treated or untreated with levothyroxine replacement) following previous total or partial thyroidectomy for multinodular goiter [50,102,122], benign single nodule [60], toxic Plummer’s nodule [124], thyroid cancer in orthotopic gland [35,42,123], thyroiditis plus benign nodules [118,120]
  • primary hypothyroidism other than congenital or iatrogenic type [37,73]”

Section 2.3.B. (thyroiditis elements in EMT patients):

“To summarize, thyroiditis profile in patients confirmed with EMT was found as following:

  • positive serum antibodies [63]
  • positive serum antibodies and pathological confirmation of thyroiditis in eutopic gland [36,118,120]
  • positive serum antiTPO and antiTg in addition to thyroiditis confirmation in both cervical thyroid and EMT [59]
  • histological report in EMT (but not in cervical eutopic thyroid) confirming focal lymphocytic thyroiditis in a 56-year-old male [105]
  • in the matter of TRAb positive status, the full-blown picture of Graves’s disease on first admission was registered in one male of 41 years that was finally identified to also have EMT [34]; another case, a 67-year old female who underwent total thyroidectomy 7 years before for non-toxic goitre, had a TRAb positive EMT [122] or previous history of Graves’s disease in one case (9 year before EMT diagnosis) [130]
  • inflammatory thyroiditis pattern in benign EMT (post-operatory confirmation) [102]
  • retrospective diagnosis of thyroiditis in eutopic thyroid due to otherwise unexplained reduced uptake amid 99m-Tc pertechnetate scintigraphy [73]”

Section 2.10. (outcome in EMTs):

“The main elements when it comes to the end results and consecutive management upon EMT identification are the followings:

  • EMT removal: based on the published data, this is the preferred approach and it is expected to have a good outcome and no local recurrence after surgery.
  • EMT surveillance: conservative approach, for instance, after EMT confirmation amid FNA results [89]) is less likely preferred.
  • Firstly, thyroid dysfunction impairs the surgical outcome thus medication with anti-thyroid drugs for newly detected hyperthyroidism [34] is mandatory and further decision to be taken during follow-up (for instance, propylthiouracil until thyroid function normalization followed by total thyroidectomy and EMT resection [121] or methimazol followed by EMT removal [122]).
  • A second surgical step after EMT resection involves a total thyroidectomy followed by radioiodine ablative therapy, TSH suppressive thyroxine treatment, and lifelong thyroglobulin monitoring in cases with differentiated thyroid cancer in EMT [35,123].
  • External beam radiation therapy was applied for bone metastases [61].
  • Thyrosine kinase inhibitors were proposed for metastatic, aggressive thyroid cancer [61].
  • Follow-up of the thyroid function after EMT removal is required in order to check for (transitory) early post-operatory hypothyroidism [73].
  • Correction of the associated (iatrogenic) hypothyroidism via thyroxine replacement is done in benign thyroid disease, too.”

Thank you

  1. In general, the paper should be properly shortened and restructured. Written in this way, it lacks focus, the data are confusedly presented, the authors fade away in the descriptions of individual previously published papers, and a significant portion of the publication is spent to simply listing the data that has already been published. This should be a review paper, and in the review paper, all previously published data should be summarized, compared, and possible differences discussed. That is a key part of any review paper, and that is missing here.

Thank you very much. We already mentioned this aspect. Thank you

  1. A simple summary lacks information about your main result (the proposed algorithm). Written in this way, it totally differs from the title of the manuscript. Please make suitable corrections and adjust the text.

Thank you very much. We adjusted the summary according to your recommendations based on the main results as followings. “Here we introduce a most complex analysis in published EMT data (N=117 patients) that identified an unexpectedly high rate of malignancy (18.8%), papillary cancer being the most frequent histological type. A rate of 5.98% amid all EMTs represented individuals confirmed with unrelated (non-thyroid) malignancies. Thyroid anomalies (other than EMT presence) were reported in 38.33% of the benign EMT, while overall malignancy rate in EMT was higher than expected according to prior data when compare to other ectopic sites”. Thank you

  1. The abstract should be shortened (sufficient data should be removed) and rewritten in a more focused way.

Thank you very much. We reduced the length of the Abstract. Thank you

  1. Figure 1 – shows that 218 full text publications were reviewed for eligibility, 70 were removed, and 89 were included in the final study. But 218-70=148. An explanation is required here.

Thank you very much. We corrected it. Thank you

  1. All abbreviations should be placed under the table.

Thank you very much. We corrected them. Thank you

  1. Improve table presentation for better readability and brevity. I recommend that you remove tables from the body text of the manuscript and provide them as a supplement. You may also present them graphically.

Thank you very much. This will be done according to the final editing. Thank you.

  1. The authors only listed previously published data (tables 1 and 2). Please make comparisons between the tables, draw conclusions, and explain them thoroughly.

Thank you very much. The aspects in malignancy are specifically highlighted in sections 2.7, as well as part of the section 3.1. Of note, this is not a head-to-head study in malignant versus non-malignancy EMT, neither this is feasible according to the current levels of statistical evidence in addition to the aim of this work and associated methods. Table 1 introduced data according to larger studies/case series, but while some of them are malignant, most of them are not. Tables 3 and 4 also introduce original data according to non-malignant EMT with/without concurrent thyroid issues. Thank you.

  1. Section 3.7 with Fig. 6 must be shifted into the results section. This is your main result.

Thank you very much. According to your recommendations we moved it to section 2.11.

Thank you

  1. Data given in its original order renders Fig. 2 (and associated lines 714-717) confusing since you refer to "our analysis" without documenting its previous performance or presentation.

Thank you very much. “Our analysis” means the results (data findings) as shown in prior subsections (including Table 2 and some data in Table 1) as following:

“To summarize, the histological types of the primary thyroid malignancies in 22 subjects (Table 1 and 2) confirmed with EMT-related cancer were as follows:

  • papillary (the most common type; the patients had any form from micro-carcinoma to severe metastatic disease); of note, one more case was introduced in prior studies-based analysis, hence, a total of eleven subjects [52]
  • follicular variant of the papillary type (N = 2)
  • Hürthle-cell thyroid follicular malignancy (N = 1)
  • poorly differentiated (N = 1)
  • anaplastic (N = 2)
  • medullary (N = 1)
  • lymphoma (N = 2)
  • MALT (mucosa-associated lymphoid tissue) (N = 1) [36,37,41,42,55,58-67,69-72].
  • an additional case of immature teratoma does not belong to the specific category of primary thyroid cancer, but it was identified in ectopic tissue (EMT) [68]”

Thank you

  1. On the other hand, some parts from the results section should be shifted into the discussion section, as they are discussing previously published data.

Thank you. They are all prior reported data and we moved some of these to Discussion according to your suggestion. Thank you

  1. The pies should be presented in a uniform way.

Thank you very much. We highlighted the sections and subsections. Also we introduced a synopsis to make it clearer. We respectfully mention that this is a massive work and we intended to cover a very large spectrum of multidisciplinary approach that is why a complex presentation is mandatory.

Thank you

  1. Please clarify and explain Figure 4— presentation, findings, and conclusions.

Thank you very much. This figure points out a “qualitative perspective of the malignancies in EMT patients (red=cancer originating from the follicular thyroid cell at any differentiation level; white=benign thyroid tissue; yellow=non-thyroid type of cancer):

1=subjects with primary cancer within the eutopic thyroid and benign EMT [22,35,123];

2=subjects with thyroid cancer in EMT and cervical thyroid (distinct foci) [36,42,55];

3=cancer in EMT, not in cervical thyroid [37,41,52,55,58,60-62,64,66,69-72];

4=malignancy in EMT with metastasis in cervical thyroid [55]; 5=prior or concurrent non-thyroid malignancies at the moment of EMT identification in terms of originating from lung [11,49,101], breast [29,92,125] or ovary [89].”

Thank you

  1. The list of references should be shortened.

Thank you very much. We respectfully mention that these data associated a wide area of medical and surgical domains and we intended to offer a complex perspective with respect to EMT. To our aware, this is the most complex analysis in EMT when compare to all the data that have been prior published and thus its length. Moreover, there are no length or references number limitations according to MDPI rules. Also, the reviewer no. 1 proposed a small expansion of the topics at Discussion that we initially did not highlight.  Thank you.

Round 2

Reviewer 1 Report

Comments and Suggestions for Authors

All comments have been thoroughly addressed. I extend my gratitude to both the authors and editors for taking my opinions into consideration during the review of this manuscript.

Author Response

Response to Review 1 Comments (second round)

Dear Reviewer,

Thank you very much for your time and your effort to review our manuscript for the second time.

We are very grateful for providing your valuable feedback on the article.

All comments have been thoroughly addressed. I extend my gratitude to both the authors and editors for taking my opinions into consideration during the review of this manuscript.

Thank you very much. We really appreciate it.

Reviewer 2 Report

Comments and Suggestions for Authors

Dear authors,

The revised version of the manuscript “Personalized management of malignant and non-malignant ectopic mediastinal thyroid; a proposed 10-item algorithm of approach," Manuscript ID: cancers-2975667, has been significantly improved. However, there are still a few minor corrections that should be implemented before it can be published:

1.     Tables should be presented as a supplement, not as part of the main content.

2.     Despite your extensive labor and diverse data, the graphical presentation of the results may benefit from simplification. This is one of the reasons that pies should be presented in a consistent manner.

3.     Figure 4 lacks clarity and is redundant. As a result, it might be eliminated from the manuscript.

4.     The final analysis includes 89 papers and there are 241 references. I still believe that the list of references is too long, and it should be reduced.

Author Response

Response to Review 2 Comments (second round)

Dear Reviewer,

Thank you very much for your time and your effort to review our manuscript for the second round.

We are very grateful for your insightful comments and observations, also, for providing your valuable feedback on the article.

Here is a point-by-point response and related amendments that have been made in the manuscript according to your review (marked in yellow color).

Dear authors,

The revised version of the manuscript “Personalized management of malignant and non-malignant ectopic mediastinal thyroid; a proposed 10-item algorithm of approach," Manuscript ID: cancers-2975667, has been significantly improved.

Thank you very much. We really appreciate it.

However, there are still a few minor corrections that should be implemented before it can be published:

Thank you very much. We mention them as followings:

  1. Tables should be presented as a supplement, not as part of the main content.

Thank you. We moved the tables, thus Table 1 became Table S1, Table 2 became Table S2, Table 3 became Table S3, Table 4 became Table S4.

  1. Despite your extensive labor and diverse data, the graphical presentation of the results may benefit from simplification. This is one of the reasons that pies should be presented in a consistent manner.

Thank you. We corrected it.

  1. Figure 4 lacks clarity and is redundant. As a result, it might be eliminated from the manuscript.

Thank you. We respectfully moved it to the end of the supplementary materials. Thank

  1. The final analysis includes 89 papers and there are 241 references. I still believe that the list of references is too long, and it should be reduced.

Thank you very much. We reduced it. Thank you
